# Visual Compositional Tuning

**Xindi Wu**[1*]   **Hee Seung Hwang**[1*]   **Polina Kirichenko**[2]   **Esin Tureci**[1]   **Olga Russakovsky**[1]

[1]Princeton University   [2]Meta AI
https://princetonvisualai.github.io/compact/

## Abstract

Visual instruction tuning (VIT) datasets have grown rapidly in scale, yet the informativeness of individual training samples has largely been overlooked. Recent dataset selection methods have shown that a small fraction of such datasets enriched with informative samples can lead to efficient finetuning of Multimodal Large Language Models. In this work, we explore the impact of sample complexity on informative data curation and introduce COMPACT (COMPositional Atomic-to-complex Visual Compositional Tuning), a visual compositional tuning data recipe that scales training sample complexity by combining multiple atomic visual capabilities in a single training example. Concretely, we synthesize rich and informative text questions for each image, allowing us to significantly reduce the number of training examples required for effective visual instruction tuning. COMPACT demonstrates superior data efficiency compared to existing data reduction methods. When applied to the LLaVA-665K VIT dataset, COMPACT reduces the data budget by 90% while still achieving 100.2% of the full VIT performance (compared to only 97.5% by the state-of-the-art method) across eight multimodal benchmarks. Further, training on the COMPACT data *outperforms* training on the full-scale VIT data on particularly complex benchmarks such as MM-Vet (+8.6%) and MMStar (+2.9%). COMPACT offers a scalable and efficient synthetic data generation recipe to improve on vision-language tasks.

## 1 Introduction

Visual instruction tuning (VIT) data for Multimodal Large Language Models (MLLMs) has continuously scaled over time. Cambrian-10M (Tong et al., 2025) is 15 times larger than LLaVA-665K (Liu et al., 2024b), and Eagle 2 (Li et al., 2025) instruction tuning data is 2.6 times larger than Cambrian-10M. As a result, state-of-the-art MLLMs like LLaVA (Liu et al., 2023; 2024a), Cambrian (Tong et al., 2025), and Eagle (Shi et al., 2024; Li et al., 2025) have shown impressive progress in a wide range of vision-language tasks (Bai et al., 2023; Alayrac et al., 2022; Li et al., 2022). The prevailing axiom of *quantity over quality* (Li et al., 2024a) in multimodal training shifted the attention of subsequent works away from the fundamental question: *Can we develop more effective data curation methods beyond blind scaling?*

Visual reasoning often relies on combining multiple fundamental visual capabilities (Ke et al., 2025; Wu et al., 2024b; Zerroug et al., 2022). Consider the question, "What color is the object on the left side of the car?". A model needs to see the car (object recognition), find what is on the left side (spatial relationship), and identify its color (color attribution) to answer the question. However, VIT datasets typically focus on individual or limited combinations of visual capabilities (e.g., "What color is the car?"), ignoring the crucial relationship between these capabilities and how they might be combined to solve complex tasks. The resulting visual reasoning questions are often simplistic and refer to a limited region in the image, under-utilizing the rich visual information within. We therefore ask whether scaling the complexity of each sample can lead to more informative datasets.

We first define a set of fundamental visual capabilities essential for visual reasoning, called *atomic capabilities* (Tab. 1), by analyzing the LLaVA-665K VIT dataset (see §3.1 and §A.4 for more details on the curation process). Using these as building blocks, we define a complexity measure, called

---

*Equal contribution

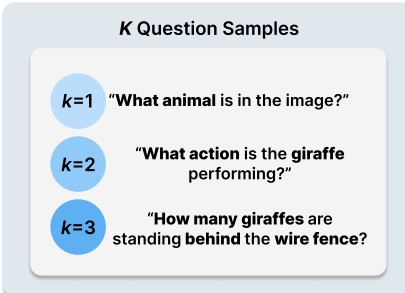 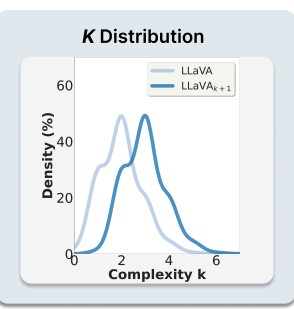 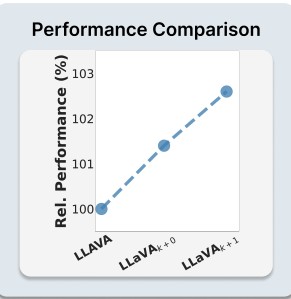

Figure 1: **Complexity $k$.** We show that increasing the complexity of LLaVA-665K improves performance. (*Left*): Examples of questions with different $k$-values, where $k$ is the number of atomic capabilities required. (*Center*): Distribution of $k$-value in VIT subset (LLaVA) and VIT subset augmented with 1 additional capability (LLaVA$_{k+1}$). Kernel density estimation is applied with bandwidth of 5. (*Right*): Performance on downstream tasks (§4.1) for VIT subset (LLaVA), VIT subset regenerated with no capability augmentation (LLaVA$_{k+0}$), and VIT subset augmented with 1 additional capability (LLaVA$_{k+1}$).

*k-value*, as the number of atomic capabilities required to answer a question. This measure allows us to quantify the complexity of each sample and identify combinations of capabilities that improve information density. For example, instead of the $k = 2$ question, "What color is the car?" (object recognition and color attribution), we can ask a $k = 3$ question, "What color is the object on the left side of the car?" (object recognition, color attribution, and spatial understanding). Analysis of existing VIT datasets reveals an over-representation of simpler ($k \leq 2$) questions that under-exploit the visual information in images.

We conduct an exploratory experiment where we take a small amount of VIT data and increase the $k$-value for a subset of examples. We first identify the atomic visual capabilities required to answer each question. We then regenerate the questions with Gemini-2.0-Flash (Team et al., 2023) after adding a randomly selected atomic visual capability. The distribution of $k$ in the dataset shifts to the right by 1, effectively increasing the average complexity of the training dataset. Fig. 1 shows that VIT on a higher $k$ dataset leads to higher downstream performance, with the generation method held constant. These results motivate our complexity-aware visual compositional tuning data recipe.

We propose **COMPACT** (COMPositional Atomic-to-complex Visual Compositional Tuning), an efficient visual compositional tuning data recipe that controls the complexity of each sample by combining atomic visual capabilities. We summarize our key contributions:

1. We show that increasing the complexity of training samples allows more effective use of information content in a given dataset. We introduce COMPACT to address limitations in complexity-agnostic scaling of conventional VIT methods.

2. We define the $k$-value to quantify the complexity of a vision-language task. We analyze the optimal distribution of complexity in the data for maximum efficiency and performance.

3. We demonstrate the effectiveness of COMPACT. With only 10% volume of the LLaVA-665K (Liu et al., 2024b) VIT dataset, training with our COMPACT data matches the performance of full-scale VIT (100.2% relative performance), and even outperforms on particularly complex multimodal benchmarks like MM-Vet (Yu et al., 2023) (29.2 when trained with full LLaVA-665K vs 31.7 when trained with COMPACT) and MMStar (Chen et al., 2024a) (35.1 vs 36.1).

## 2 RELATED WORK

**Visual Instruction Tuning.** Instruction following is an essential capability in language models (Wei et al., 2021; Zhou et al., 2023). Misalignment between a model's response and the format requested by a question can hinder the precise evaluation of its performance (He et al., 2024; Hsieh et al., 2023; Salido et al., 2025; Balepur et al., 2025). VIT involves training a model on a fixed set of response patterns that can be repeated during inference (e.g., multiple-choice, short- and long-response questions) (Liu et al., 2023; 2024b). Although VIT has shown performance improvements

Table 1: **Taxonomy of atomic capabilities.** We identify 10 atomic capabilities that are necessary for visual reasoning. We categorize them into attribution, recognition, and relation. Atomic capabilities serve as building blocks for building complex tasks.

| Group | Capability | Definition | Example Question |
|---|---|---|---|
| **Attribution** | Color | Identifying or comparing colors of objects | What **color** is the car? |
| | Shape | Recognizing and describing shapes of objects | What **shape** is the dining table? |
| **Recognition** | Object Recognition | Identifying and naming objects present | What **object** is on the table? |
| | Action Recognition | Identifying what action is being performed | What is the person **doing**? |
| | Text Recognition | Reading and interpreting text visible | What **word** is written on the sign? |
| | Counting | Determining the number of instances | **How many** cars are there? |
| | Spatial Recognition | Understanding a scene's overall layout | Which object is **closest** to the viewer? |
| **Relation** | Spatial Relationship | Locating objects relative to each other | What is **next to** the red car? |
| | Object Interaction | Analyzing how multiple objects interact | What is the dog **chasing**? |
| | Scene Understanding | Identifying the type of environment/setting | **Where** is this scene taking place? |

in general multimodal capabilities (Huang et al., 2023), recent work (Ghosh et al., 2024) has shown that optimizing for response formatting potentially limits the quality of language model responses.

While VIT (Liu et al., 2024b) relies on simple questions to train instruction-following capability, COMPACT leverages more complex questions to achieve the same goal. Higher $k$ questions facilitate learning by encouraging the model to learn from more visual features in each image.

**Complex Tasks in LLMs and MLLMs.** Complexity in the space of vision-language tasks has been loosely conceptualized by different interpretations of compositionality. Some studies view compositionality as a sequential arrangement of basic tasks (Chen et al., 2024b; Li et al., 2024c). Compositionality as integrations of basic capabilities has been examined in geometric reasoning (Chae et al.) and general visual reasoning (Hua et al., 2024), but mainly in the context of evaluation.

We argue that compositional complexity can be studied in the context of training MLLMs. Studies highlight that while MLLMs do show signs of compositional capability (Ossowski et al., 2024), they struggle when constituting components and their combined patterns are not strongly learned or missing during training (Campbell et al., 2025). COMPACT builds on this insight by defining complexity as the combination of atomic visual capabilities and widening the range of complexity represented in the dataset.

**Scaling Visual Instruction Tuning Data.** VIT is a data- and compute-heavy step in training (Xu et al., 2023). Prior work has approached this challenge mainly from two directions. First, upscaling-based approaches (Tong et al., 2025; Liu et al., 2024b) argue that ever larger corpora can continue to improve visual capabilities, partially addressing the challenge of low information density through sheer volume of data. Second, data selection studies argue that full performance can be reproduced with smaller amounts of data (Lee et al., 2024; Liu et al., 2024c). ICONS (Wu et al., 2024a) shows that models can achieve near-full performance across a suite of MLLM benchmarks using only a fraction of the original VIT dataset.

However, these approaches treat complexity as a byproduct of scale rather than a controllable metric. Our visual compositional tuning approach, COMPACT, exposes the model to more complex tasks without scaling the dataset size by explicitly targeting higher $k$-values during sample generation.

## 3 METHOD

We aim to construct a training dataset in which each sample requires knowledge of the fundamental visual capabilities and their meaningful combinations that progressively increase the complexity of the task. Inspired by the compositional nature of visual reasoning, we first define complexity as a metric that arises from combining basic visual capabilities. This requires identification of a set of fundamental visual capabilities that can be combined, called atomic visual capabilities (§3.1). A subset of these capabilities is combined to generate a new sample with target complexity, which corresponds to COMPACT's four-step data recipe (§3.2).

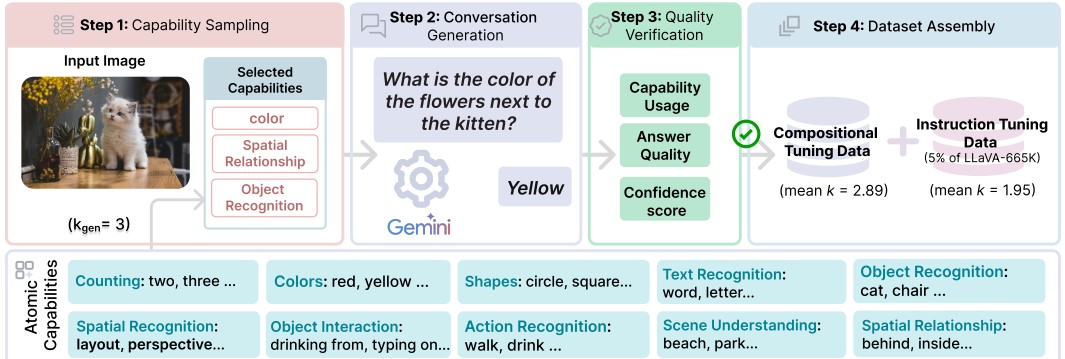

Figure 2: **COMPACT data generation pipeline.** *(Left):* We design a data recipe that can scale the complexity of each training example. We randomly sample $k_{gen} \in \{1, 2, 3\}$ atomic capabilities such as color, object recognition, and spatial relationship. *(Center):* We generate questions that integrate all $k_{gen}$ sampled capabilities and verify their quality. *(Right):* We combine the synthetically generated compositional tuning data with instruction tuning data for response formatting.

## 3.1 ATOMIC VISUAL CAPABILITIES

Atomic capabilities are foundational skills that can be combined to solve complex tasks. For example, a model needs to acquire object recognition, color attribution, and spatial relationship understanding capabilities to identify how objects of different colors are spatially oriented. For each task $T$, we identify a set of atomic visual capabilities $\{c_1, \ldots, c_k\}$ required to solve this task. We define the number of atomic capabilities required to solve the task $T$ as its *complexity k*.

We build the taxonomy of atomic capabilities (Tab. 1) based on two key considerations. First, we focus on vision-centric skills that can encourage the model to utilize the visual information in the image. Therefore, we exclude non-perceptual capabilities (e.g., cultural knowledge, historical context, coding, and math) that require external information beyond the image. Second, we include capabilities that can also be found in existing VIT data curation methods (Liu et al., 2023; Tong et al., 2025; Li et al., 2025), for fair comparison of our approach to others. As a result, we create a list of 10 atomic capabilities which can be divided into three major categories: 1) **Attribution**: identifying visual properties (e.g., color and shape). 2) **Recognition**: identifying visual entities, including objects, actions, and text. 3) **Relation**: identifying visual interactions and spatial orientations.

We note that the atomic capabilities in Tab. 1 are not expected to cover the entire multimodal task space or be completely orthogonal (see Fig. 8 in the Appendix). Rather, we aim to find sufficiently distinct capabilities that allow meaningful combinations to generate diverse visual reasoning tasks. We provide more details on the taxonomy curation in Appendix §A.4.

## 3.2 VISUAL COMPOSITIONAL TUNING DATA RECIPE

Visual compositional tuning scales the complexity of training data by composing multiple atomic visual capabilities into a single sample. COMPACT generates multi-capability questions $\mathcal{D}_{comp}$ by prompting vision-language models to create questions that require natural[1] integration of multiple atomic visual capabilities. This process involves four key steps: (1) sampling a subset of capabilities from our predefined set of atomic visual capabilities (**Capability Sampling**), (2) prompting `Gemini-2.0-Flash` (Team et al., 2023) to generate questions that integrate all the selected capabilities (**Conversation Generation**), (3) validating the capability requirement and the quality of each question (**Quality Verification**), and (4) combining our generated compositional tuning data with a small portion of the LLaVA-665K (Liu et al., 2024b) data for response formatting (**Dataset Assembly**).

**Step 1: Capability Sampling.** We start by taking a random sample of images from LLaVA-665K (Liu et al., 2024b). For each image, we uniformly sample $k_{gen} \in \{1, 2, 3\}$ capabilities from

---

[1]We use the term "natural" to denote a combination of visual capabilities that correspond to their co-occurrence patterns in real-world settings, wherein multiple capabilities are integrated in a way that is contextually and semantically meaningful.

our predefined pool of 10 atomic visual capabilities. At this stage, $k_{gen}$ functionally serves as the lower bound of the actual $k$-value of the generated conversation. We sample $k_{gen}$ with the expectation that the final $k$-value of the sample will eventually be higher ($k_{gen} \leq k$), as some atomic capabilities are weakly correlated in practice (see Fig. 8 in the Appendix). To maximize capability coverage, we do the following in each round of capability sampling: (a) prioritize the capabilities that have not been selected for that image, and (b) drop duplicate combinations of capabilities for the same image. These efforts ensure that our training examples capture diverse visual information.

**Step 2: Conversation Generation.** For each capability combination that is sampled, we prompt `Gemini-2.0-Flash` (Team et al., 2023) to generate a conversational question-answer pair that integrates all capabilities in the combination, as well as a confidence score between 0 and 100 that represents its confidence in the quality of the conversation. Our carefully designed prompt (see Appendix §B) enforces several key constraints: (a) questions must require the use of visual information from the image and cannot be answered from its text alone, (b) answers must be concise, (c) questions must integrate the specified capabilities naturally (without using conjunctions to simply conjoin single-capability questions), and (d) questions must reference objects and features actually present in the image. The purpose of these constraints is to produce vision-centric conversations that are unambiguous and natural.

**Step 3: Quality Verification.** We include a verification process with `Gemini-2.0-Flash` (Team et al., 2023) to ensure the quality and diversity of the training dataset. We filter out questions with uninformative answers (e.g., "unknown", "not visible") or those with confidence scores below 70%. We discard questions that share more than 60% of the words with those previously accepted.

Since the generator may not always successfully integrate all sampled capabilities into a single question, we perform capability verification by prompting `Gemini-2.0-Flash` (Team et al., 2023) to analyze whether each generated question indeed requires the $k_{gen}$ specified capabilities (see Appendix §B). Steps 2 and 3 repeat iteratively until we collect 2-3 high-quality conversations per $k_{gen}$ for each image or reach a maximum of 10 attempts.

**Step 4: Dataset Assembly.** We address the challenge of aligning the response format by mixing our synthetically generated compositional tuning data with a random 5% subset of the LLaVA-665K (Liu et al., 2024b) VIT dataset. This mixture of instruction tuning and compositional tuning data has the following effects. First, the VIT subset maintains the model's ability to handle diverse response formats and instructions required by modern MLLM benchmarks (e.g., multiple-choice questions (Fu et al., 2023), open-ended answers (Liu et al., 2024b)). Second, our compositional tuning data leverages complex questions to facilitate learning with higher information density. In this way, we delegate the instruction-following capability training to the original VIT data and allow our compositional tuning data to focus on visual reasoning.

The size of the compositional tuning data is determined by the minimum number of images needed to match full LLaVA-665K (Liu et al., 2024b) VIT performance. COMPACT preserves the contents of the images, which enables us to fairly compare against existing methods in their ability to extract rich visual information.

## 4 EXPERIMENTS

In this section, we evaluate the baseline approaches and our visual compositional tuning method, COMPACT, on existing multimodal benchmarks. First, we discuss our evaluation setup and the benchmarks (§4.1). Second, we compare the performance of COMPACT with relevant baselines, including LLaVA-665K (Liu et al., 2024b) VIT (§4.2). Third, we investigate the source of COMPACT's performance improvement (§4.3). Finally, we ablate the various design choices of COMPACT (§4.4).

### 4.1 EVALUATION TESTBED

**Model.** We train the LLaVA-v1.5-7B-LoRA (Liu et al., 2024b) model's pre-visual-instruction-tuning checkpoint[2] on our COMPACT training dataset. This checkpoint has not been exposed to any

---

[2]LLaVA-v1.5-mlp2x-336px-pretrain-vicuna-7b-v1.5, which has no prior exposure to visual instruction tuning data.

Table 2: **Baseline comparisons.** COMPACT outperforms baseline VIT data recipes on multimodal benchmarks. With only 5% of the LLaVA-665K (Liu et al., 2024b) VIT data and 32K of our compositional tuning data (65K total), COMPACT outperforms the random subset of the VIT data (Random), various data reduction methods, as well as the full VIT data. The best and second-best results for each benchmark are shown in **bold** and underlined, respectively. We further provide results using an open-source generator (Qwen3-VL-4B-Instruct) for COMPACT in Tab. 10.

| Recipe | # Data | InfoVQA | SeedBench2Plus | MME | TextVQA | MM-Vet | CV-Bench | MMStar | LLaVA-W | Rel. (%) |
|---|---|---|---|---|---|---|---|---|---|---|
| LLaVA-665K | 665K | 20.80 | 41.72 | **1478.48** | **46.99** | 29.22 | **60.92** | 35.11 | **68.50** | 100.00 |
| Random | 65K | 20.05 | 41.85 | 1327.70 | 42.88 | 30.46 | 54.71 | 34.13 | 64.30 | 95.38 |
| EL2N | 65K | 20.52 | 42.95 | 1378.58 | 42.41 | **33.53** | 50.92 | 33.82 | 66.40 | 97.09 |
| Perplexity | 65K | 20.46 | 41.90 | 1375.32 | 42.95 | 30.32 | 52.72 | 33.47 | 68.40 | 96.09 |
| SemDeDup | 65K | 20.54 | **43.96** | 1431.36 | 42.71 | 28.39 | 42.24 | 34.18 | 62.10 | 93.31 |
| D2-Pruning | 65K | 20.90 | 43.70 | 1343.30 | 41.82 | 31.61 | 48.49 | **36.63** | 68.40 | 97.13 |
| Self-Sup | 65K | 20.61 | 42.51 | 1434.30 | 42.68 | 30.18 | 54.30 | 34.33 | 61.20 | 96.04 |
| Self-Filter | 65K | 19.99 | 41.33 | 1290.34 | 36.59 | 28.21 | 45.17 | 33.67 | 66.40 | 90.51 |
| ICONS | 65K | 21.0 | 42.03 | 1402.75 | 43.12 | 31.23 | 55.96 | 35.96 | 61.8 | 97.47 |
| COMPACT (ours) | 65K | **23.68** | 43.13 | 1379.94 | 44.37 | 31.74 | 55.28 | 36.13 | 64.50 | **100.18** |

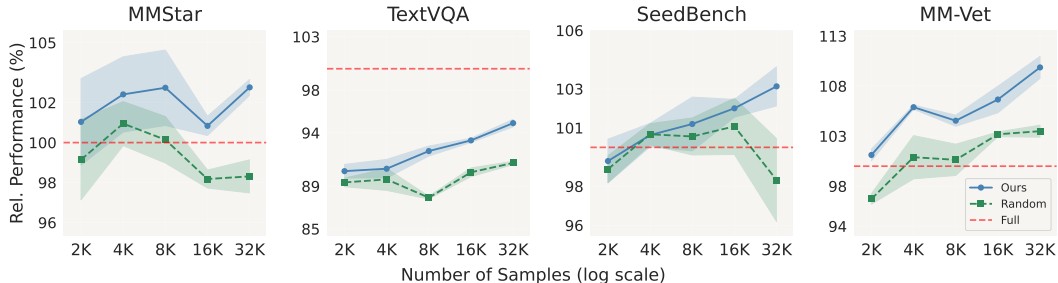

Figure 3: **Performance across visual compositional tuning data scales.** We show that COMPACT's compositional tuning data scales more efficiently than conventional VIT. We fix the VIT subset (5% of LLaVA-665K (Liu et al., 2024b)) and scale the compositional tuning data in COMPACT from 2K to 32K. We compare each mix with VIT-only datasets with equal data budgets. COMPACT (solid lines) consistently outperforms LLaVA-665K VIT (dashed lines) with less data. COMPACT's compositional tuning data scales particularly well on SeedBench2Plus (Li et al., 2024b), which consists of spatially complex tasks of navigating charts and maps. Shaded regions indicate ±1 standard deviation across 3 runs.

visual instruction tuning data prior to COMPACT training. The training dataset includes 32K-sample compositional tuning data unless otherwise stated. Additionally, we mix 5% of LLaVA-665K (Liu et al., 2024b) to preserve instruction-following capability. We train the model for one epoch with its official LLaVA-v1.5 LoRA fine-tuning settings.

**Baselines.** We compare the effectiveness of our COMPACT data recipe with several baseline datasets by training models with the same architecture under identical training configurations. **LLaVA-665K**: The full LLaVA-665K (Liu et al., 2024b) VIT dataset (665K samples) used in LLaVA-v1.5. This serves as our primary performance baseline. **Random**: A 65K-sample random subset of LLaVA-665K that matches our training data size. This baseline controls for data volume. We further evaluate 65K-sample subsets curated with various data reduction methods: EL2N (Paul et al., 2021), Perplexity (Marion et al., 2023), SemDeDup (Abbas et al., 2023), D2-Pruning (Maharana et al., 2023), Self-Sup (Sorscher et al., 2022), Self-Filter (Chen et al., 2024c), and ICONS (Wu et al., 2024a).

**Benchmarks.** We evaluate models trained with different data recipes on established multimodal benchmarks that assess complex visual capabilities. 1) **MM-Vet** (Yu et al., 2023) includes 16 types of complex multimodal tasks integrated from 6 core capabilities (recognition, OCR, knowledge, language generation, spatial awareness, and math). 2) **MME** (Fu et al., 2023) contains 10 perception (e.g., color, count, OCR) and 4 cognition (e.g., commonsense reasoning, text translation, code understanding) related visual subtasks. 3) **LLaVA-in-the-Wild** (Liu et al., 2024b) is an open-ended visual question answering benchmark that asks complex questions on real-world images. 4) **SeedBench2Plus** (Li et al., 2024b) evaluates visual comprehension skills of MLLMs with a focus on charts, maps, and webs. 5) **MMStar** (Chen et al., 2024a) contains 1,500 visual questions that span 6 core capabilities (fine-grained perception, coarse perception, mathematics, science &

Table 3: **Matching LLᴀVA-665K distribution.** We show that performance improvements of COMPACT can be attributed to increasing complexity $k$. COMPACT$_{llava}$ capability distribution exactly matches that of the random LLᴀVA-665K subset, controlling for complexity. Note that we use 16K compositional data (different from 32K in Tab. 2) for faster iteration combined with 5% of the LLᴀVA-665K (Liu et al., 2024b) VIT data. Roughly half of the performance gain in COMPACT can be attributed to the higher-$k$ compositional tuning data.

| Recipe | #Data | InfoVQA | SeedBench2Plus | MME | TextVQA | MM-Vet | CV-Bench | MMStar | LLaVA-W | Rel. (%) |
|---|---|---|---|---|---|---|---|---|---|---|
| LLᴀVA-665K | 665K | 20.80 | 41.72 | 1478.48 | 46.99 | 29.22 | 60.92 | 35.11 | 68.50 | 100.00 |
| Random | 49K | 20.33 | 42.38 | 1290.45 | 42.22 | 30.18 | 54.75 | 34.3 | 70.5 | 96.28 |
| COMPACT$_{llava}$ | 49K | 22.76 | 43.43 | 1308.52 | 43.94 | 28.81 | 52.39 | 36.08 | 66.8 | 97.55 |
| COMPACT | 49K | 22.68 | 42.82 | 1362.68 | 43.73 | 30.78 | 54.69 | 35.59 | 66.6 | 98.83 |

technology, logical reasoning, and instance reasoning), carefully curated to evaluate multimodal understanding. 6) **CV-Bench** (Tong et al., 2025) is a MLLM benchmark specialized for 2D and 3D visual understanding that includes spatial relationship, object count, relative distance, and depth order. 7) **TextVQA** (Singh et al., 2019) evaluates visual understanding of texts in the image. 8) **InfoVQA** (Mathew et al., 2022) measures visual understanding of infographic images. These benchmarks cover a broad range of vision-centric capabilities. We also note that some of these benchmarks include non-visual questions involving such skills as knowledge and math, which are not our primary focus. We provide detailed discussion of these tasks in Appendix §A.2 and §A.3.

## 4.2 Main Results

**Overall Performance.** As shown in Tab. 2, COMPACT performs on par with the LLᴀVA-665K (Liu et al., 2024b) baseline with only 10% of its data volume. COMPACT's training dataset contains a mixture of 32K compositional tuning data and 5% of the LLᴀVA-665K (Liu et al., 2024b) VIT data (33K). The compositional tuning data trains the model on complex questions, and the VIT subset maintains the model's instruction-following capability.

Across all benchmarks, our COMPACT achieves an average relative performance of 100.2%, outperforming the full LLᴀVA-665K (Liu et al., 2024b) (100.0%), random baseline (95.4%), and ICONS (Wu et al., 2024a) (97.5%). COMPACT achieves strong gains on tasks like MM-Vet (Yu et al., 2023) (+8.6% over LLᴀVA-665K (Liu et al., 2024b)), MMStar (Chen et al., 2024a) (+2.9%), InfoVQA (Mathew et al., 2022) (+13.8%), and SeedBench2Plus (Li et al., 2024b) (+3.4%) while maintaining competitive performance on TextVQA (Singh et al., 2019) and LLaVA-in-the-Wild (Liu et al., 2024b). This highlights the effectiveness of the COMPACT data recipe. Additionally, we provide qualitative results in Appendix §C.

**Complex Training Data is Efficient.** We study the data efficiency of complex samples by analyzing how COMPACT's performance changes as we scale the amount of compositional tuning data. We fix the VIT subset (5% of LLᴀVA-665K (Liu et al., 2024b)) and scale the compositional tuning data in COMPACT from 2K to 32K. For comparison, we match the size of each dataset purely with VIT data. Fig. 3 shows that as the number of compositional tuning data samples increases, the performance of COMPACT on multimodal benchmarks steadily improves, unlike the random baseline. Interestingly, models trained on a smaller amount of compositional tuning data (2K-8K samples) often match or exceed the performance of random baseline models trained on much larger VIT data. For instance, COMPACT's 2K model achieves 30.8 on MM-Vet (Yu et al., 2023), outperforming the random baseline's 32K model at 30.5. This demonstrates that COMPACT makes more effective use of training data compared to the baselines.

This improvement in data efficiency comes from several factors. COMPACT training data has higher-$k$ samples that encourage the model to extract more visual information from the image. The learning potential of lower-$k$ samples that LLᴀVA-665K (Liu et al., 2024b) relies on is limited by the amount of visual information that the model can afford to ignore. However, a range of simple-to-complex training samples is still necessary to properly disentangle each atomic capability from complex questions. COMPACT's complexity-aware data generation process that progressively increases the complexity of the samples balances these effects.

### 4.3 ANALYSIS

**Complexity Distribution in LLAVA-665K.** We characterize the complexity distribution of the LLAVA-665K (Liu et al., 2024b) VIT dataset that serves as the primary baseline for COMPACT. We use `Gemini-2.0-Flash` (Team et al., 2023) to analyze the $k$-value of 5,400 questions that belong to 1,000 randomly sampled images (see the details of the system prompt in Appendix §B). Fig. 1 shows that the mean $k$-value of the samples is approximately $k = 1.95$, and the mode is $k = 2$. 77% of the questions require two or fewer atomic visual capabilities. Interestingly, a small fraction of the questions (0.06%) require as many as 10 capabilities (e.g., "Question: Describe this photo in detail."). We also observe that 0.9% of the questions require zero capabilities. We further provide $k = 0$ examples (see Appendix §C.2).

**Complexity Distribution in COMPACT.** We confirm that COMPACT increases the complexity distribution of the training dataset previously established by LLAVA-665K. We use `Gemini-2.0-Flash` (Team et al., 2023) to analyze 7,200 questions that belong to 1,000 randomly sampled images in the compositional tuning data. The mean $k$-value of the data is $k = 2.89$, and the mode is $k = 3$, which are higher than the LLaVA counterparts. We also find that the representation of each capability in the dataset is more balanced compared to LLAVA-665K (see Appendix Fig. 6). We provide more details on the data statistics of COMPACT and LLAVA-665K data in Appendix §A.5.

COMPACT successfully scales the $k$-value distribution of the training dataset. The number of capabilities sampled during generation ($k_{gen}$) in the **Capability Sampling** step (§3.2) practically acts as the lower bound of the question's final $k$-value for two reasons: 1) The object recognition capability is implicitly assumed by the conversation generator (i.e., Gemini (Team et al., 2023)) during generation and verification steps (see Fig. 7 in the Appendix). We hypothesize that grounding questions in specific objects is implicitly required to generate high-quality questions about the image, likely due to biases in the image sources. We note that COMPACT does not change the visual contents of the images. 2) We find that spatial capabilities such as scene understanding, spatial recognition, and spatial relationship are often related in practice as the question becomes spatially complex (see Fig. 8 in the Appendix). The spatial orientation of objects can be influenced by the overall layout of the scene. For example, what is "on the left side of" an object could be described as "behind", depending on their relative depth.

**Effect of Matching LLAVA-665K Distribution.** We isolate the complexity effect from COMPACT's performance gain by controlling for the sample generator. We generate 16K-sample compositional tuning data whose capability and $k$-value distribution matches that of LLAVA-665K (Liu et al., 2024b). Similar to the original COMPACT data recipe, we mix this complexity-matched compositional tuning data with the random 5% subset of the VIT data. We compare this training dataset with the following baselines: 1) a random subset of the VIT data with the same size, 2) original COMPACT with 16K compositional tuning data, and 3) the full VIT dataset. As shown in Tab. 3, the performance of complexity-matched COMPACT (COMPACT$_{llava}$) stands at 97.6% (compared to the full VIT baseline). The performance of original COMPACT jumps to 98.8%, suggesting that at least half of the performance gain (1.3% out of 2.6% improvement over random baseline) in COMPACT comes from higher complexity in the compositional tuning data.

### 4.4 ABLATION STUDIES

We conduct ablation studies to understand how COMPACT addresses the aforementioned challenges in using complexity as a tool to control sampling for VIT data curation. First, we vary the range of complexity in the data to show that training efficiency benefits from higher and wider ranges of $k$. Second, we show that response formatting can be trained by mixing a relatively small amount of prompt-engineered VIT dataset. Unless otherwise specified, all experiments use 5% of LLAVA-665K (Liu et al., 2024b) VIT data and 16K compositional tuning data.

**Effect of $k$-value Range.** We analyze the relationship between information density and efficient learning based on the learning potential of different ranges of complexity. We train the model on COMPACT with five different versions of 16K compositional tuning data, each generated using $k_{gen} = 1$, $k_{gen} = 2$, $k_{gen} = 3$, $k_{gen} = 1, 2$, or $k_{gen} = 1, 2, 3$ in the **Capability Sampling** step (§3.2). For fair comparison, we maintain consistent sample counts and use an identical set of images in all

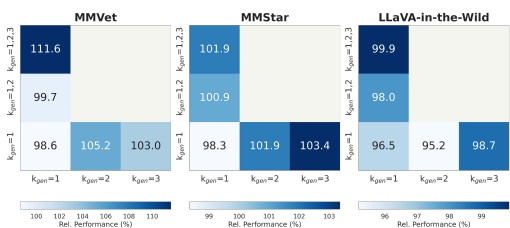

Figure 4: **Impact of k-value range:** Performance comparison across variations of COMPACT whose compositional tuning data is synthesized with different ranges of $k_{gen}$ in the **Capability Sampling** step (§3.2). Models trained on higher and wider ranges of complexity (darker bars) achieve higher performance.

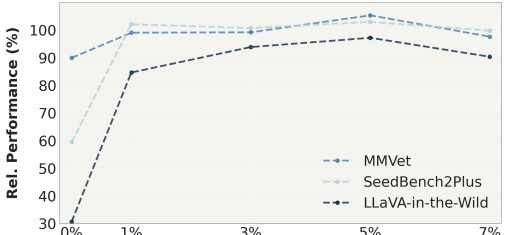

Figure 5: **Impact of instruction tuning data ratio.** Relative performance of COMPACT with different amounts of instruction tuning data from LLaVA-665K (Liu et al., 2024b). The x-axis is the percentage of LLaVA-665K used as instruction tuning data, and the y-axis is relative score. The performance improves significantly with a small amount of instruction tuning data and stabilizes around 5%.

five settings. As shown in Fig. 4, increasing the number of sampled capabilities per question leads to consistent improvements on all three benchmarks. Training on $k_{gen} = 1, 2, 3$ compositional tuning data achieves the highest performance on MM-Vet (Yu et al., 2023) and LLaVA-in-the-Wild (Liu et al., 2024b), and the second-highest on MMStar (Chen et al., 2024a). Surprisingly, sampling from $k_{gen} = 1, 2, 3$ demonstrates stronger performance on average compared to sampling only $k_{gen} = 3$. This suggests that although complex samples are more information dense, their benefit is maximized in the presence of simpler samples. We provide more details on k-value range in Appendix §A.2.

**Impact of Instruction Tuning Data Ratio.** We show that the domain adaptation problem of fitting the model's response to the format required by the test dataset can be resolved with a data mixture. We fix the 16K compositional tuning data and scale the amount of the VIT subset from 0% (pure visual compositional tuning) to 7% of LLaVA-665K (Liu et al., 2024b). Fig. 5 shows that a small addition of the VIT subset improves the performance of the model on MM-Vet (Yu et al., 2023) (short answer), SeedBench2Plus (Li et al., 2024b) (multiple choice), and LLaVA-in-the-Wild (Liu et al., 2024b) (long answer). We find that the prompt-engineered VIT subset trains the instruction-following capability of the model on various question types even at just 1%. Interestingly, further scaling gives diminishing returns and starts to lose absolute gains at 7%. These results suggest that instruction-following capability is potentially orthogonal to the capabilities of the base model and the atomic visual capabilities, and can be acquired with minimal instruction tuning data.

## 5 DISCUSSION

**Conclusion.** In this work, we introduce COMPACT, a visual compositional tuning data recipe that increases the complexity of training samples by combining a higher number of atomic visual capabilities (e.g., object recognition, spatial reasoning, shape attribution). Our experimental results show that training on more complex samples matches the full LLaVA-665K (Liu et al., 2024b) VIT performance across benchmarks with less than 10% of the original data budget. Our work demonstrates visual compositional tuning as a scalable, data-efficient pathway toward multimodal models that can solve multi-capability tasks.

**Limitations.** Our approach faces two key limitations. First, we mainly rely on data generated from closed-source models (i.e., Gemini (Team et al., 2023)), which potentially introduce their compositional limitations and biases to our dataset. We provide further discussion on failure modes in Appendix §A.6. Additionally, this data generation process is costly, which could pose challenges for reproducibility. Second, our approach focuses on the compositionality of vision-centric capabilities. Therefore, our approach may not be optimal for addressing knowledge-intensive tasks that lie outside the scope of visual reasoning. A detailed discussion on knowledge-intensive task is in Appendix §A.3.

**Future Work.** We aim to extend visual compositional tuning by investigating the intersection of image complexity and text complexity. Currently, our data recipe generates questions given a fixed set of images. Specifically, certain groups of atomic visual capabilities and complexity levels might be optimal for different types of visual content present in the image. Additionally, investigating how complexity impacts learning efficiency of different learning algorithms such as reinforcement learning would be promising avenues of future research.

**Acknowledgments.** This material is based upon work supported by the National Science Foundation under Grants 2107048 and 2112562, and the Solidigm AI SW. Any opinions, findings, and conclusions, or recommendations expressed in this material are those of the author(s) and do not necessarily reflect the views of the National Science Foundation and Solidigm Research. All experiments, data collection, and processing activities were conducted at Princeton University. Meta was involved solely in an advisory role and no experiments, data collection or processing activities were conducted on Meta infrastructure. We thank Allison Chen, Sanghyuk Chun, and Jihoon Chung for helpful discussions and feedback.

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

# Appendices

# A    ADDITIONAL ANALYSIS

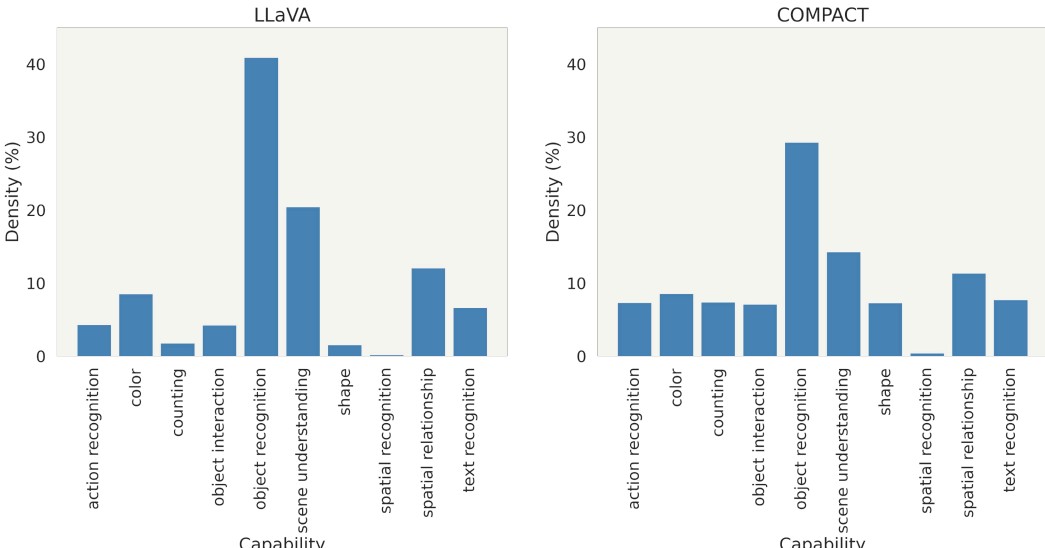

Figure 6: **Comparison of capability distribution.** The bar plots show the frequency of each atomic capability in LLaVA (*left*) and COMPACT (*right*) samples. In LLaVA, the distribution is notably imbalanced: object recognition and scene understanding are some of the most frequent, while shape and spatial recognition are less prevalent. In contrast, COMPACT exhibits a more balanced distribution across capability categories.

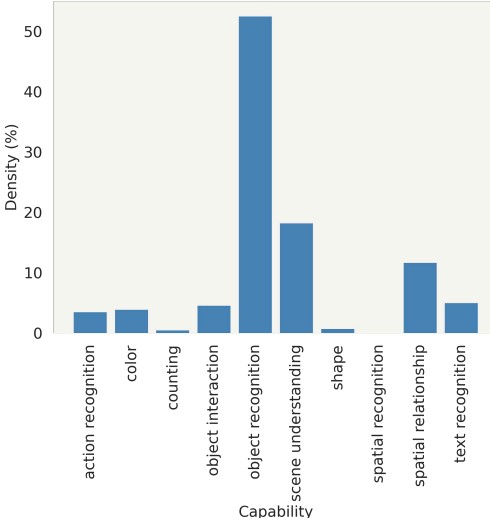

Figure 7: **Distribution of implicitly assumed capabilities.** The bar plot shows the frequency of atomic capabilities that are implicitly assumed during COMPACT's compositional tuning data generation. Object recognition is most commonly assumed during question generation, most likely because the images are object-centered.

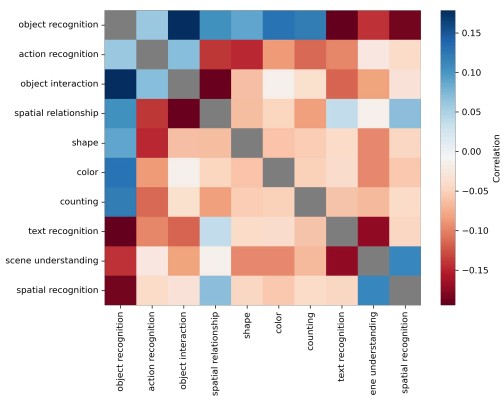

Figure 8: **Correlation between capabilities.** The heatmap shows the correlation between unique capabilities in COMPACT's compositional tuning data. Object recognition's correlations with other capabilities are relatively strong. Spatial capabilities are also locally correlated, as spatial questions require understanding of the scene, depth, and relative position in practice.

Table 4: **Limited performance improvements on knowledge-intensive benchmarks.** Comparison shows modest improvements over random baseline on tasks that require substantial world knowledge or domain expertise. Numbers reported in accuracy (%) and relative performance to full model (%).

| Recipe | OK-VQA | MMMU | MMMU-Pro | | **Rel.** |
| | | | Standard | Vision | (Avg.) |
| --- | --- | --- | --- | --- | --- |
| Random | 49.30 | 32.89 | 18.15 | 11.44 | 92.0% |
| COMPACT | 50.02 | 33.89 | 20.23 | 11.91 | 96.6% |
| LLaVA-665K (Liu et al., 2024b) | 57.96 | 33.89 | 20.12 | 11.97 | 100% |

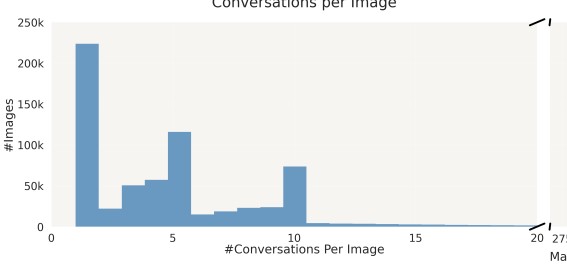

Figure 9: **Distribution of conversations per image in LLaVA-665K.** The overwhelming majority of images (97.69%) have $\leq 20$ conversation pairs. The average number of conversations per image is 5.18 ($\sigma = 5.62$). A small subset (2.31%) exceeds 20 conversations, which includes a sample with the maximum length of 275. Total conversations: 3,444,246.

Table 5: **TextVQA performance by k-value.** Performance of LLaVA-v1.5-7b on each k-value. $k \geq 4$ is excluded as it contains fewer than 10 samples. Approximately 5% of the questions are $k = 0$, which are ambiguous and do not strictly refer to the image.

| | k = 1 | k = 2 | k = 3 |
| --- | --- | --- | --- |
| Density | 73.6% | 19.1% | 2.2% |
| LLaVA-v1.5-7B | 46.4 | 45.6 | 39.8 |

A.1 RELATIONSHIP BETWEEN K-VALUE AND COMPLEXITY OF THE TASK

We use TextVQA (Singh et al., 2019) to demonstrate that increasing the k-value makes the task progressively more challenging. Every question in this benchmark contains one key atomic capability (text recognition) in addition to potentially others. We split the TextVQA validation set according to the k-value of each question, and report the results of the LLaVA-v1.5-7b model.

Tab. 5 shows that the LLaVA-v1.5-7b model struggles as more capabilities (e.g., color, spatial relation understanding) are added to text recognition. This provides further evidence that complexity measured by the k-value is related to the model's ability to process visual information, and that the k-value characterizes the complexity of the question in a meaningful way. In short, as the k-value increases, overall task difficulty increases.

We report on the TextVQA benchmark because the results are more interpretable. Every question requires a shared atomic capability (text recognition) and has the same answer format, allowing us to isolate the effect of adding capabilities more strictly.

A.2 BREAKDOWN OF PERFORMANCE ON MMSTAR

**Category-specific performance.** We split the questions in MMStar (Chen et al., 2024a) by category (provided by the dataset) and report category-specific results for the models. Tab. 6 shows that COMPACT improves performance across most categories except math, which we exclude from our list of atomic capabilities for fair comparison with training on LLaVA-665K data. We observe particularly large improvements on science & technology (diagram and chart understanding) and instance reasoning (perception and relation understanding) categories, suggesting that COMPACT broadly generalizes to visual tasks.

**k-value specific performance.** We provide further analysis to explain the mechanism of improvement in COMPACT. We split the questions in MMStar by k-value and measure the performance of three models, each trained on three different datasets ($k_{gen} = 1$, $k_{gen} = 1, 2$, and $k_{gen} = 1, 2, 3$; Fig. 4) with progressively higher k-value distributions. Tab. 7 shows that $k_{gen} = 1, 2, 3$ outperforms others on higher k-value ($k \geq 2$) questions. The improvement is larger when we compare $k_{gen} = 1, 2, 3$ and $k_{gen} = 1$, as opposed to $k_{gen} = 1, 2, 3$ and

Table 6: **Category-specific performance on MMStar.** Performance of COMPACT and the random LLaVA-665K subset on each category in MMStar. COMPACT data has 33K VIT data and 16K compositional tuning data.

| MMStar | Data | coarse perception | fine-grained perception | instance reasoning | logical reasoning | math | science and technology |
|--------|------|-------------------|-------------------------|--------------------|--------------------|------|------------------------|
| Random | 49K | 59.5 | 28.9 | 38.1 | 29.9 | 28.7 | 25.2 |
| COMPACT | 49K | 61.6 | 28.9 | 41.2 | 31.8 | 28.1 | 27.7 |
| Improvement | | +3.5% | 0% | +8.1% | +6.4% | -2.1% | +9.9% |

Table 7: **k-value specific performance on MMStar.** Performance improvements on each k-value by $k_{gen} = 1, 2, 3$ dataset compared to $k_{gen} = 1$ and $k_{gen} = 1, 2$.

| MMStar | Data | k = 1 | k = 2 | k = 3 | k = 4 |
|--------|------|-------|-------|-------|-------|
| $k_{gen} = 1, 2, 3$ vs $k_{gen} = 1$ | 49K | -0.5% | +3.6% | +22.7% | +33.5% |
| $k_{gen} = 1, 2, 3$ vs $k_{gen} = 1, 2$ | 49K | -1.3% | +2.6% | +14.1% | +9.1% |

Table 8: **Atomic capability ablation.** Average drop in relative performance as a result of ablating each atomic capability during 16K compositional tuning data generation in COMPACT.

| Atomic Capability | Rel. Drop | Atomic Capability | Rel. Drop |
|-------------------|-----------|-------------------|-----------|
| w/o scene understanding | -5.2% | w/o counting | -3.3% |
| w/o spatial relationship | -4.9% | w/o object interaction | -3.2% |
| w/o text recognition | -4.7% | w/o spatial recognition | -3.1% |
| w/o object recognition | -4.0% | w/o action recognition | -2.1% |
| w/o color | -3.7% | w/o shape | -0.7% |

$k_{gen} = 1, 2$. These results suggest that increasing the k-value of the training data enables the model to perform well on higher k-value questions in the test dataset.

Tab. 7 also shows that training on $k_{gen} = 1, 2, 3$ leads to a small drop in performance on lower k-value ($k = 1$) questions, indicating a trade-off between lower and higher k-value regimes. This explains why both simple and complex samples are necessary for training.

## A.3 PERFORMANCE ON KNOWLEDGE-INTENSIVE TASKS

While our visual compositional tuning recipe shows general improvements on various benchmarks, we observe more modest gains on knowledge-intensive tasks. Tab. 4 compares the performance of different approaches on OK-VQA, MMMU, and MMMU-Pro benchmarks. COMPACT with 32K compositional tuning data shows relatively small improvements over the random baseline: OK-VQA (50.02% vs 49.30%), MMMU (33.89% vs 32.89%), and MMMU-Pro (20.23% vs 18.15% on standard tasks, 11.91% vs 11.44% on vision tasks). Notably, training on the full LLaVA-665K VIT dataset leads to limited performance improvements on MMMU (33.89%). Although knowledge-related tasks are not our main focus, this inspires future work on designing visual compositional tuning approaches that cover broader capabilities outside of the vision space.

## A.4 ATOMIC CAPABILITY SELECTION

We conduct ablations on each atomic capability to analyze which capability drives the most gains in COMPACT. For each iteration, we prepare an ablated COMPACT training dataset after excluding each atomic capability in the data generation step (Step 2). Tab. 8 shows the average drop in relative performance across multimodal benchmarks (§4.1) when we ablate each atomic capability. The results indicate that all atomic capabilities contribute non-trivially to COMPACT's performance.

**Atomic Capability Selection.** The original LLaVA-665K data curation process lays out 8 of the 10 atomic capabilities in COMPACT; we identify the two others (object interaction and shape attribution) manually from inspecting the dataset. We acknowledge that alternative taxonomies are possible depending on specific training goals, and we view our 10 atomic capabilities as a practical starting point rather than a definitive set.

Table 9: **Token-level comparison.** Comparison between COMPACT and LLaVA datasets (32k entries each).

| Metric | COMPACT | LLaVA | Difference |
|---|---|---|---|
| Input tokens (mean) | $12.83 \pm 4.18$ | $16.85 \pm 25.49$ | 31% shorter |
| Output tokens (mean) | $1.70 \pm 0.90$ | $21.74 \pm 73.38$ | 92% shorter |
| Tokens per Q&A turn | 14.53 | 38.59 | 62% fewer |
| Total tokens per entry | 104.87 | 197.42 | 46.88% reduction |

**Non-Orthogonality of Capabilities.** We acknowledge that perfect orthogonality is neither achievable nor necessary for our framework. Real-world visual reasoning naturally involves overlapping skills, for example, spatial reasoning often co-occurs with object recognition. The correlation analysis in Fig. 8 reveals these natural dependencies, which we view as a feature rather than a limitation. The key insight is that combining multiple capabilities, even correlated ones, requires the model to integrate more visual information, which increases information density.

**Implicit Assumption of Object Recognition.** Fig. 7 shows that object recognition is frequently implicitly assumed during question generation. Object recognition serves as a foundational capability that often appears alongside other skills, similar to how reading comprehension is fundamental to many NLP tasks. Rather than undermining decomposability, this reflects the hierarchical nature of visual reasoning where basic perception enables higher-level reasoning. This foundational role is consistent with the design of vision-centric instruction tuning, where grounding responses in specific objects is essential for meaningful visual understanding.

## A.5   Analysis on Data Statistics

**Conversation length distribution in LLaVA-665K.** Fig. 9 shows the distribution of the number of conversations per image in LLaVA-665K (Liu et al., 2024b). 93.6% of the samples fall below the 10-pair threshold. The distribution's mean of 5.18 conversations per image ($\sigma = 5.62$) shows that the data is heavily skewed toward lower values. We fix the target number of conversations per image in the compositional tuning dataset based on these findings. We ensure a fair comparison by aligning the distribution of our data with the baseline distribution.

**Token-level analysis on COMPACT data.** We conduct a token-level analysis comparing our COMPACT-generated compositional tuning data (32K samples) with an equivalent-sized LLaVA-665K subset (32K samples). The results demonstrate that COMPACT achieves substantial token efficiency despite incorporating multiple atomic questions. Tab. 9 reveals that COMPACT uses 104.87 tokens/sample (3.36M total tokens) compared to LLaVA's 197.42 tokens/sample (6.32M total tokens), achieving a 46.88% reduction (92.55 fewer tokens per entry). Specifically, COMPACT's input (question) tokens average $12.83 \pm 4.18$ (31% shorter than LLaVA's $16.85 \pm 25.49$), and answer tokens average $1.70 \pm 0.90$ (92% shorter than LLaVA's $21.74 \pm 73.38$). The token reduction translates directly to faster training ($\sim$47% fewer tokens to process), lower storage usage (shorter sequences), and better data efficiency (more focused atomic capabilities per token).

## A.6   Quality Verification and Failure Mode Analysis

We conduct a systematic analysis of failure modes in our generated compositional tuning data to ensure quality and transparency. To quantify the effectiveness of our quality control process, we perform a controlled experiment generating questions for $k_{gen} = 1, 2, 3$ on a sample of images.

**Rejection Rates and Failure Modes.** Our multi-stage filtering (described in §3.2) rejected approximately 21% of generated questions across four primary failure modes: (1) **Low confidence scores (10%)**: questions where the VLM cannot answer confidently, often due to ambiguous phrasing or requiring information not present in the image; (2) **Uninformative responses (12.5%)**: questions with answers like "unknown", "not visible", "yes", or "no" that do not provide meaningful visual supervision (for example, "What text is visible on the wooden tray?" answered with "None"); (3) **High word overlap/near-duplicates (40%)**: questions that share more than 60% of words with previously accepted questions for the same image (for instance, generating both "What is the color of the bench in the image?" and "What is the color of the bench located in the center of the scene?" with 70% word overlap); and (4) **Capability mismatch (37.5%)**: questions that do not naturally integrate the specified capabilities (for example, asking "What object is present in the image without any action being performed?" for $k_{gen} = 2$ with action_recognition and object_recognition, where the question only requires object recognition, as the VLM correctly identified only 1 of the 2 expected capabilities). Notably, capability mismatch becomes increasingly important as $k$ increases, accounting for 0%, 50%, and 66.7% of rejections for

Table 10: **Baseline comparisons (Qwen3 generator).** Following Tab. 2, we report COMPACT results using an open-source generator (Qwen3-VL-4B-Instruct), along with the full LLaVA-665K (Liu et al., 2024b) baseline and our compact COMPACT recipe for reference.

| Recipe | # Data | InfoQA | SeedBench2Plus | MME | TextVQA | MM-Vet | CV-Bench | MMStar | LLaVA-W | Rel. (%) |
|---|---|---|---|---|---|---|---|---|---|---|
| LLaVA-665K | 665K | 20.80 | 41.72 | 1478.48 | 46.99 | 29.22 | 60.92 | 35.11 | 68.50 | 100.00 |
| COMPACT (ours, Gemini) | 65K | 23.68 | 43.13 | 1379.94 | 44.37 | 31.74 | 55.28 | 36.13 | 64.50 | 100.18 |
| COMPACT (ours, Qwen3) | 65K | 22.97 | 42.12 | 1370.76 | 43.07 | 30.78 | 56.38 | 34.04 | 65.70 | 98.31 |

$k_{gen} = 1, 2, 3$, respectively. This demonstrates that our verification successfully filters questions that artificially force capability combinations.

**Common Failure Patterns in Remaining Data.** Despite our rigorous filtering, some failure patterns remain in the dataset: (1) **Overly complex $k_{gen} = 3$ questions**: some high-complexity questions are difficult even for humans to answer, potentially due to the challenge of naturally integrating three or more capabilities in a single question; (2) **VLM misidentification errors**: cases where the VLM generator misidentifies image content, leading to incorrect ground-truth answers (while our verification step mitigates this issue, it does not eliminate it entirely); (3) **Spatial reasoning errors**: questions involving spatial relationships when objects are ambiguously positioned, leading to potential disagreement about the correct answers; and (4) **Attribute questions for occluded objects**: color or shape questions about small or partially occluded objects where the attributes are not clearly visible. However, COMPACT achieves consistent improvements across diverse benchmarks with different evaluation protocols, suggesting these biases do not critically harm generalization. This multi-stage quality control ensures that our training data maintains both high quality and natural capability integration.

## A.7 Additional Experiments with Open-source Data Generator

We evaluate COMPACT using an open-source generator (Qwen3-VL-4B-Instruct) to test whether our gains persist without proprietary generation. Tab. 10 reports these results, alongside the full LLaVA-665K baseline and our compact COMPACT recipe for direct comparison.

## B  SYSTEM PROMPTS

**System Prompts for Sample Generation and Capability Analysis.**    We provide the system prompts for compositional question generation (**A**) and verification (**B**). The generation prompt includes structured guidelines to ensure that the generated multi-capability questions naturally blend different capabilities and can only be answered by checking the corresponding images. The verification prompt checks if the questions meet these guidelines and do not contain subjective interpretations or compositional flaws. We also provide the system prompt for our capability analysis (**C**) where we identify all the required capabilities for a given question.

For conversation generation, we use Gemini-2.0-Flash (Team et al., 2023) with temperature 0.1, top-p 0.9, max token 1000. We have 3 runs per question on average. We use 32 parallel processes to generate 32K compositional tuning data. Each generation uses about 700 tokens on average. The total generation time is roughly 2 hours and the API cost is $86.5.

---

### (A-1) System Prompt for Question Generation: Guidelines and Capability Definitions

**Prompt**: You are an AI assistant that generates challenging but well-defined questions and answers about images. First, I will provide you with k specific capabilities. Generate 1 question that naturally integrates EXACTLY these k capabilities.
**IMPORTANT**:

- If the question can be answered without looking at the image (e.g., the answer can be inferred from the question itself or previous questions), it's a BAD question
- Questions should be reasonably challenging but must have clear, unambiguous answers
- All answers must be extremely concise - use only a single word or short phrase
- Each question must be a single, integrated question that naturally combines all k given capabilities
- DO NOT use "and" or commas to combine separate questions
- Questions should require careful observation and reasoning
- Only generate questions when you can determine the answer with high confidence
- Avoid subjective or ambiguous questions
- ONLY ask about objects and capabilities that are ACTUALLY PRESENT in the image
- NEVER create questions about objects or features that don't exist in the image
- Generate diverse questions that differ in topic and required reasoning

**CAPABILITY DEFINITIONS**:

- spatial_relationship: Identifying how specific objects are positioned relative to each other (above, below, next to, inside, etc.) - focuses on the direct relationship between two or more particular objects
- spatial_recognition: Understanding the overall spatial layout and arrangement of the entire scene - focuses on the general organization, depth, perspective, or environmental context, rather than relationships between specific objects
- text_recognition: Reading and interpreting text visible in the image
- action_recognition: Identifying what action is being performed (can involve a single person/object)
- object_interaction: Analyzing how multiple objects interact with each other (requires at least two objects) - MUST involve at least one moving/active object, not just static objects positioned together - can include humans interacting with objects and humans interacting with humans
- object_recognition: Identifying and naming objects present in the image
- counting: Determining the number of instances of something in the image
- color: Identifying or comparing colors of objects in the image
- shape: Recognizing and describing the shapes of objects in the image
- scene_understanding: Identifying where the image is taken or the type of environment/setting (indoor/outdoor, beach, mountain, kitchen, office, etc.) - focuses on identifying the overall scene, background, or context of the image

---

## (A-2) System Prompt for Question Generation: Examples

**Examples**:

- **BAD**: "What color is the car, and where is it located?" (two separate questions)
- **BAD**: "What might the person be thinking?" (subjective/ambiguous)
- **BAD**: "Is this a nice room?" (subjective)
- **BAD**: "What breed of dog is in the corner?" (when no dog exists in the image)
- **BAD**: "How are the fridge and desk interacting?" (static objects don't qualify as interaction)
- **BAD**: "What is the color of the red car?" (answer can be inferred from the question itself without seeing the image)
- **GOOD**: "What color car is parked next to the red brick building?" (specific, clear answer)
- **GOOD**: "How many yellow tennis balls are visible on the wooden court?" (requires counting + color)
- **GOOD**: "What is the person in blue using to interact with the television?" (proper object interaction)
- **GOOD**: "Where is this image taken?" (scene understanding)
- **GOOD**: "Where is this scene happening?" (scene understanding)

## (B) System Prompt for Question Verification

**Prompt**: You are an AI assistant that verifies if questions about images properly utilize specified capabilities.
Given a question and its answer, analyze whether it NATURALLY requires using EXACTLY k specified capabilities - no more, no less.
**IMPORTANT**:

- The question should require ALL specified capabilities to be answered
- The question should not require additional major capabilities beyond those specified
- The capabilities must be naturally integrated, not artificially forced

## (C) System Prompt for Capability Analysis

**Prompt**: You are an AI assistant that analyzes questions to identify the core capabilities required to answer them.
Given a question, identify ALL the capabilities it requires from this list:
- spatial relationship (understanding relative positions)
- object interaction (how objects/people interact)
- scene understanding (understanding the background)
- text recognition (reading text in images)
- spatial recognition (understanding 3D space)
- action recognition (identifying actions/activities)
- object recognition (identifying objects)
- counting (counting objects/people)
- color (identifying colors)
- shape (identifying shapes)

Return ONLY a JSON array of the required capabilities, like: ["capability1", "capability2"]

## C  VISUALIZATIONS

### C.1  QUALITATIVE COMPARISON

We provide qualitative visualizations that compare the outputs from our compositionally-tuned COMPACT model and the LLAVA-665K VIT model. Examples in Fig. 10 highlight the importance of visual compositional tuning for handling complex multi-capability tasks ($k \geq 3$). These cases demonstrate the COMPACT model's enhanced ability to integrate multiple visual capabilities, while showing the baseline model's difficulty with such compositionally complex queries.

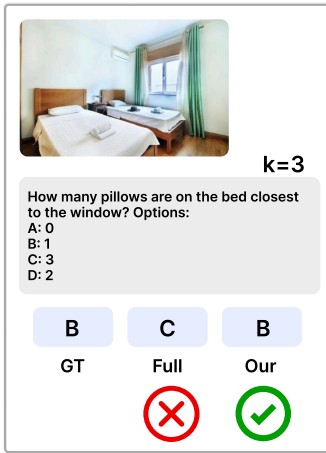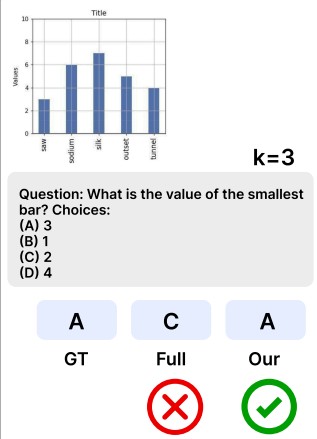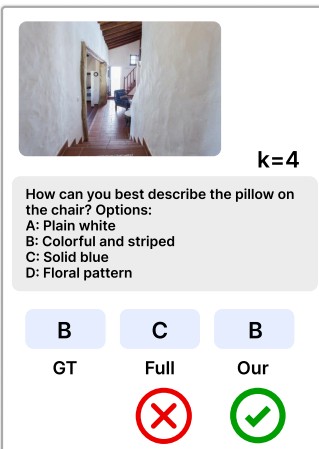

Figure 10: **Qualitative comparison of model outputs.** Examples showing responses from our compositionally-tuned COMPACT model and LLAVA-665K (Liu et al., 2024b) VIT model on complex queries that require multiple capabilities ($k \geq 3$). Our model demonstrates better integration of visual capabilities which leads to more accurate responses.

## C.2 Zero-Capability Samples in LLaVA-665K

We identify a subset of samples in the LLaVA-665K dataset that requires no visual capabilities, which we refer to as zero-capability samples. These include general knowledge queries, subjective prompts, or requests that can be answered without inspecting the image at all. While such data may still be useful for instruction following, it does not contribute to the development of vision-centric skills. In our analysis, we find that approximately 1.1% of the questions in LLaVA-665K fall into this zero-capability category.

---

**Zero-Capability Samples**

**Zero-Capability Questions**:

- How is the weather?
- Should I move to London?
- Can you provide some information about the Emirates airline?
- Give me a long list of what duties are considered rental activity
- Have the cat declare her new name as ruler
- rewrite it from the perspective of an expensive therapist
- Can you tell me how to prepare a Colombian dish
- how to do coding
- Can you explain Map Reduce to me?
- A 35 year old patient presented to the emergency department with shortness of breath. Before this, he was at a crowded event. He does not have a history of diabetes or high blood pressure. He had a positive PCR test at an outside hospital. What should be the next steps for the physician?
- please convert those snomed codes to FHIR
- I'm running a used car dealership, what are some emerging opportunities for me brought by large language models like GPT-3?
- answer it again in Chinese
- you are a legislator. You are asked to come up with a framework for new legislation that adances the science of reading for grades K-3. Write that model legislation.
- I'm looking to create a podcast, can you help me?

---

## C.3 COMPACT Data Visualization

We visualize the COMPACT dataset to provide insights into its compositional structure. Figs. 11 and 12 show selected examples from the COMPACT dataset. Each question is generated from a combination of $k$ atomic capabilities. These cases demonstrate our model's enhanced ability to integrate multiple visual capabilities simultaneously, while the baseline model often struggles with such compositionally complex queries.

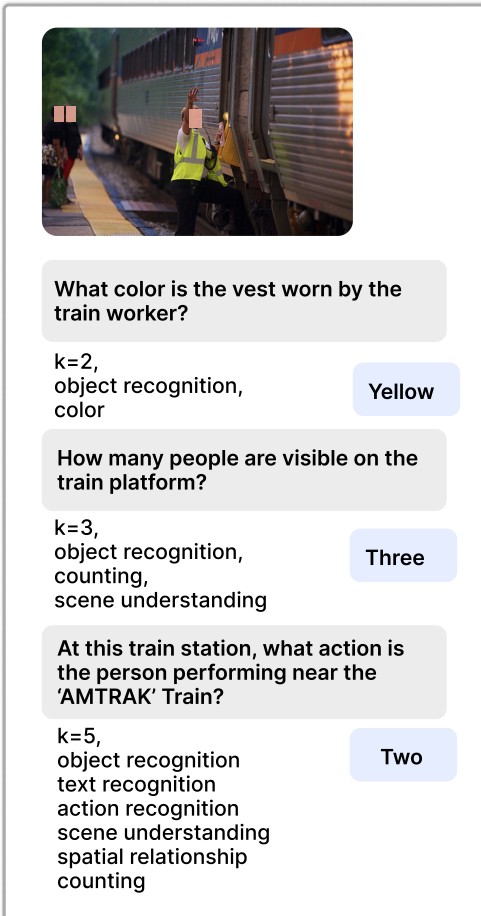
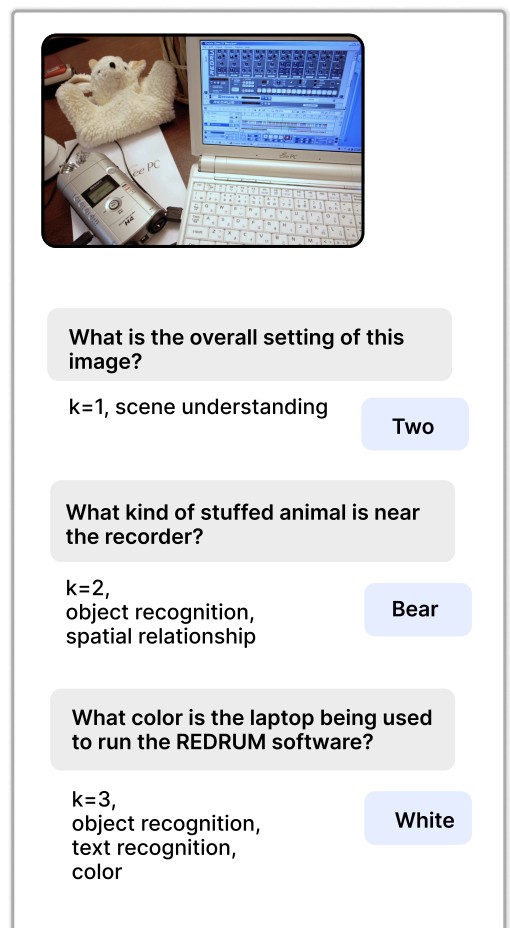

Figure 11: **Visualization of COMPACT visual compositional tuning samples.**

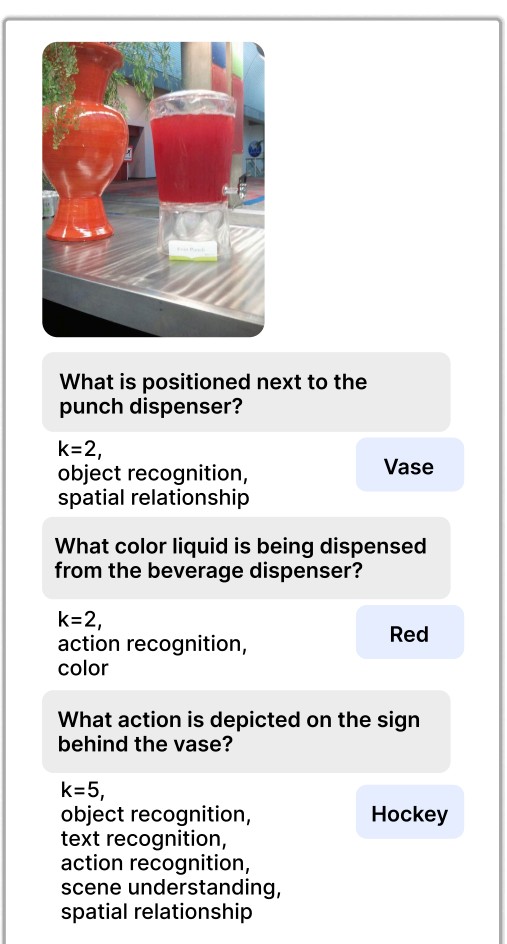

**What is positioned next to the punch dispenser?**

k=2,
object recognition,
spatial relationship

Vase

**What color liquid is being dispensed from the beverage dispenser?**

k=2,
action recognition,
color

Red

**What action is depicted on the sign behind the vase?**

k=5,
object recognition,
text recognition,
action recognition,
scene understanding,
spatial relationship

Hockey

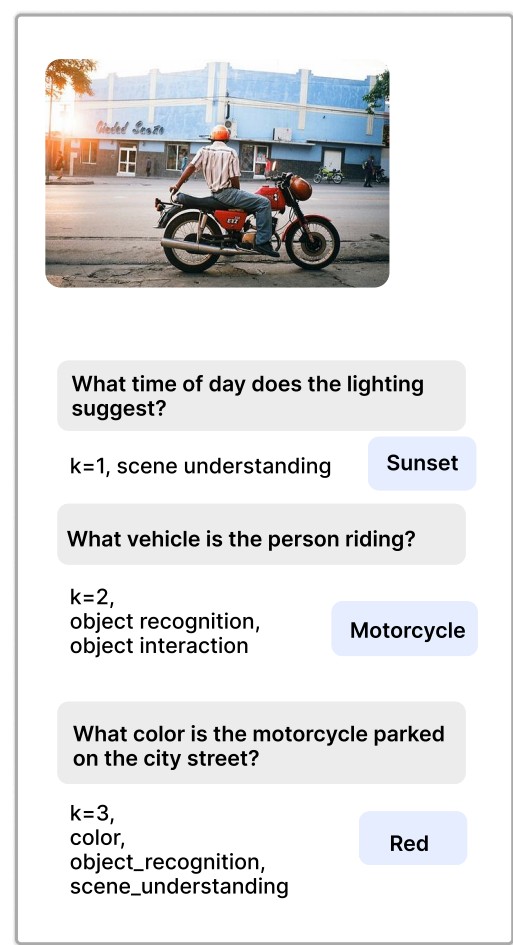

**What time of day does the lighting suggest?**

k=1, scene understanding

Sunset

**What vehicle is the person riding?**

k=2,
object recognition,
object interaction

Motorcycle

**What color is the motorcycle parked on the city street?**

k=3,
color,
object_recognition,
scene_understanding

Red

Figure 12: **Visualization of COMPACT visual compositional tuning samples.**

