# OpenReview forum: "Visual Compositional Tuning"
_ICLR.cc/2026/Conference — ICLR 2026 Poster_

### Official Review · Reviewer_YsZe · 2025-10-31

**Soundness:** 3
**Presentation:** 3
**Contribution:** 3
**Rating:** 6
**Confidence:** 3

**Summary:**

This paper presents COMPACT, a data-efficient visual instruction tuning (VIT) framework that synthesizes training samples with controlled compositional complexity. The authors introduce the k-value, representing the number of atomic visual capabilities (e.g., object recognition, spatial reasoning) required to answer a question. By generating high-k samples using Gemini-2.0-Flash and combining them with a small subset of LLaVA-665K for instruction formatting, COMPACT attains 100.2% of the full-dataset performance using only 10% of the data. It substantially outperforms baselines on complex benchmarks such as MM-Vet (+8.6%) and MMStar (+2.9%).

Key contributions:
1. A complexity-aware VIT data recipe leveraging atomic capability composition.
2. Empirical evidence that higher-k samples enhance data efficiency.
3. A scalable synthetic data generation framework that reduces dependence on large-scale datasets.

**Strengths:**

1. Originality:
COMPACT introduces a novel lens for understanding and curating VIT data by framing compositional complexity through the k-value, which quantifies the number of atomic visual capabilities (e.g., object recognition, spatial reasoning, counting) required to solve a task. This provides a principled and measurable axis for dataset construction, enabling systematic control over the difficulty and diversity of visual-instruction samples—an aspect largely overlooked in prior work.

2. Quality:
The study demonstrates strong methodological rigor, featuring comprehensive evaluations across eight multimodal benchmarks and detailed ablation analyses. These experiments confirm the method’s robustness, showing consistent improvements in both generalization and compositional reasoning while maintaining competitive performance under limited data budgets.

3. Clarity:
The paper presents a well-structured taxonomy of atomic visual capabilities and clearly articulates the data synthesis pipeline, from capability composition to instruction formatting. The process is transparent, reproducible, and amenable to scaling, offering valuable guidance for future research in efficient multimodal data generation.

4. Significance:
By achieving superior performance with only 32K synthetic samples, surpassing models trained on over 665K human-annotated examples, COMPACT directly challenges the prevailing assumption that larger datasets are always necessary for stronger multimodal performance. This redefines the data–efficiency frontier and highlights compositional control as a promising new direction for scaling visual instruction tuning.

**Weaknesses:**

1. ​​Dependency on closed-source models:​​ Data synthesis via Gemini-2.0-Flash risks reproducibility and may inherit model biases. Experiments with open-source generators would strengthen generalizability.
2. ​​Limited scope of atomic capabilities:​​ Non-visual skills (e.g., knowledge, math) are excluded, limiting gains on benchmarks like OK-VQA. Expanding the taxonomy could broaden applicability.
3. ​​Evaluation of compositional generalization:​​ While high-ksamples improve performance, tests for zero-shot compositionality are lacking.

**Questions:**

1. How might COMPACT perform if atomic capabilities are expanded to include non-perceptual skills (e.g., commonsense reasoning)? Could this address the modest gains on knowledge-intensive tasks?
2. Have you explored generating data with open-source VLMs to assess reproducibility and reduce reliance on Gemini?
3. Does the benefits of high-ksamples diminish beyond k=3? Is there an optimal complexity ceiling for efficient learning?
4. How does COMPACT handle images where certain atomic capabilities are inherently hard to combine?

---

> ### Author Response · Authors · 2025-11-27
>
> **W1, Q2.** We conducted additional experiments using the open-source Qwen3-VL-4B-Instruct model as an alternative data generator. As shown in table below, COMPACT (ours, qwen3) with 65K samples achieves 98.31\% relative performance compared to the full LLaVA-665K dataset, demonstrating strong results that are competitive with our Gemini-based approach (100.18\%). The results show consistent improvements over random baseline (95.38\%) across most benchmarks, with notable gains on InfoVQA (22.97 vs 20.05), SeedBench2Plus (42.12 vs 41.85), and LLaVA-W (65.70 vs 64.30). This shows that the benefit of curating data with COMPACT is not specific to Gemini but robust across different generators.
>
> | Recipe | # Data | InfoVQA | SeedBench2Plus | MME | TextVQA | MM-Vet | CV-Bench | MMStar | LLaVA-W | Rel. (%) |
> |--------|--------|---------|----------------|-----|---------|--------|----------|--------|---------|----------|
> | LLaVA-665K | 665K | 20.80 | 41.72 | **1478.48** | **46.99** | 29.22 | **60.92** | 35.11 | **68.50** | 100.00 |
> | Random | 65K | 20.05 | 41.85 | 1327.70 | 42.88 | 30.46 | 54.71 | 34.13 | 64.30 | 95.38 |
> | EL2N | 65K | 20.52 | 42.95 | 1378.58 | 42.41 | **33.53** | 50.92 | 33.82 | 66.40 | 97.09 |
> | Perplexity | 65K | 20.46 | 41.90 | 1375.32 | 42.95 | 30.32 | 52.72 | 33.47 | 68.40 | 96.09 |
> | SemDeDup | 65K | 20.54 | **43.96** | 1431.36 | 42.71 | 28.39 | 42.24 | 34.18 | 62.10 | 93.31 |
> | D2-Pruning | 65K | 20.90 | 43.70 | 1343.34 | 41.82 | 31.61 | 48.49 | **36.63** | 68.40 | 97.13 |
> | Self-Sup | 65K | 20.61 | 42.51 | 1434.30 | 42.68 | 30.18 | 54.30 | 34.33 | 61.20 | 96.04 |
> | Self-Filter | 65K | 19.99 | 41.33 | 1290.34 | 36.59 | 28.21 | 45.17 | 33.67 | 66.40 | 90.51 |
> | ICONS | 65K | 21.0 | 42.03 | 1402.75 | 43.12 | 31.23 | 55.96 | 35.96 | 61.8 | 97.47 |
> | COMPACT (ours) | 65K | **23.68** | 43.13 | 1379.94 | 44.37 | 31.74 | 55.28 | 36.13 | 64.50 | **100.18** |
> | COMPACT (ours, qwen3) | 65K | 22.97 | 42.12 | 1370.76 | 43.07 | 30.78 | 56.38 | 34.04 | 65.70 | 98.31 |
>
> **W2, Q1.** This is an excellent suggestion for future work. Our focus on vision-centric capabilities is motivated by the specific role of VIT in the MLLM training pipeline. VIT data, including LLaVA-665K which serves as our baseline, is designed to train models to effectively utilize visual information and ground responses in image content. In contrast, knowledge-intensive capabilities (e.g., world knowledge, commonsense reasoning) are primarily learned during pre-training through large-scale text corpora. Given this context, COMPACT's emphasis on perceptual capabilities aligns with the core objective of VIT: instruction following grounded in visual content. By focusing on vision-centric skills, COMPACT encourages models to extract and compose rich visual information from images. Including knowledge-intensive capabilities would make it difficult to isolate the effect of compositional complexity on visual learning efficiency.
>
> However, expanding the taxonomy to include non-perceptual skills is possible, as the selection of atomic capabilities is independent from COMPACT's compositional data generation framework. Moreover, our category-specific analysis on MMStar in the table below shows that COMPACT improves performance on logical reasoning (+6.4\%) and science \& technology tasks (+9.9\%), suggesting that stronger visual grounding transfers to tasks requiring reasoning beyond pure perception.
>
> | MMStar | Data | coarse perception | fine-grained perception | instance reasoning | logical reasoning | math | science and technology |
> |--------|------|-------------------|-------------------------|--------------------|--------------------|------|------------------------|
> | Random | 49K | 59.5 | 28.9 | 38.1 | 29.9 | 28.7 | 25.2 |
> | COMPACT | 49K | 61.6 | 28.9 | 41.2 | 31.8 | 28.1 | 27.7 |
> | Improvement | | +3.5\% | 0\% | +8.1\% | +6.4\% | -2.1\% | +9.9\% |
>
> These results demonstrate that stronger perceptual grounding can transfer to tasks requiring reasoning beyond pure perception. Expanding the taxonomy to include non-perceptual capabilities could further improve performance on knowledge-intensive benchmarks like OK-VQA and MMMU, which we view as promising future work.

---

> > ### Author Response · Authors · 2025-11-27
> >
> > **W3, Q3.** Thank you for your question. We noticed that increasing k indefinitely (beyond $k \geq 4$) does lead to diminishing returns for two main reasons. Combining higher number of atomic capabilities poses a challenge for the data generator (e.g. Gemini) to naturally combine all atomic capabilities (ln. 238 in the paper), especially because the capabilities are sampled uniformly. Furthermore, analysis on the benchmarks shows that the average k-values are $\bar{k} < 4$. We find that sampling $k_{gen}=1,2,3$ capabilities during generation strikes the right balance between leveraging complexity for better learning and the quality of the generated samples.
> >
> > **Q4.** Our quality verification step (Step 3 in the data generation pipeline) addresses this concern. When generating questions for a given image with target k-value, we use the VLM to verify that the question can be naturally answered from the image content with high confidence (70\%). If the generated question requires capabilities that are not present or cannot be naturally combined in the image (e.g., asking about color when the image is grayscale, or spatial relations when only one object is present), the verification step rejects the sample. This ensures that all training samples maintain high quality and natural capability integration.

---

### Official Review · Reviewer_jUJz · 2025-10-31

**Soundness:** 4
**Presentation:** 3
**Contribution:** 3
**Rating:** 6
**Confidence:** 4

**Summary:**

This paper focuses on the curation of informative training data to enhance MLLMs’ finetuning efficiency. It introduces COMPACT, a novel data synthesis approach that generates rich and informative text questions for each image by integrating multiple atomic visual capabilities into a single training sample. Experimental results across various benchmarks demonstrate that COMPACT significantly reduce the required number of training examples while achieving performance comparable to that of full-scale training data, highlighting its efficiency.

**Strengths:**

Strength:
1. This work defines atomic capabilities essential for general visual reasoning and introduces the k-value metric to quantify task complexity.
2. The proposed COMPACT scales training sample complexity by incorporating multiple atomic visual capabilities within a single data, revealing that increased complexity enhances information utilization.
3. Experiments show the effectiveness of COMPACT, which achieves comparable or even superior performance with just 10% of the training data, improving finetunig efficiency for MLLMs.

**Weaknesses:**

1. The entire generation, verification, and evaluation process relies on the closed-source Gemini model, which may introduce potential bias and limit reproducibility.
2. The exploration is confined to the LLaVA-v1.5-7B-LoRA model and the LLaVA-665K VIT dataset, leaving the performance of COMPACT with other models and training datasets underexplored, especially considering that the LLaVA-665K dataset exhibits relatively low task complexity.
3. There is a lack of comparison with other data reduction methods in experiments.

**Questions:**

1. Although the number of required training data decreases, does the incorporation of multiple atomic questions in a single COMPACT question imply that the token count for inputs and outputs hasn't reduced such significantly?
2. In Figure 3, why do models trained on random data sometimes outperform those trained on the full dataset? Additionally, why does the notably poor performance of COMPACT on the TextVQA dataset?
3. What is the task complexity of the evaluation benchmarks?

---

> ### Author Response · Authors · 2025-11-27
>
> **W1.** We conducted experiments with Qwen3-VL-4B-Instruct as an alternative open-source data generator. As shown in the table below, COMPACT (ours, qwen3) achieves 98.31\% relative performance compared to the full LLaVA-665K dataset, which is competitive with our Gemini-based approach (100.18\%) and substantially better than random baseline (95.38\%). The consistent improvements across both closed-source (Gemini-2.0-Flash) and open-source (Qwen3-VL) generators suggest that COMPACT is robust to generator choice.
> We also clarify that for evaluation, we use GPT-4o as the judge model for benchmarks that require LLM-based evaluation following standard practices. We emphasize that the core contribution of COMPACT is leveraging the k-value to control sampling rather than any specific model dependency. We will release all generated data, prompts, and code to ensure full reproducibility.
>
>
> **W2.** We appreciate this feedback. We chose the LLaVA-665K VIT dataset and LLaVA-v1.5-7B-LoRA model as our primary experiment setup as LLaVA-665K is one of the most widely adopted open-source VIT datasets, making our results directly comparable to a large body of prior work. Alternative VIT datasets and models have been continuously scaling (ln. 033 in the paper) or are closed-source, making it difficult to conduct rigorous and flexible analysis. To demonstrate that COMPACT generalizes beyond this setting, we conducted additional experiments with Qwen2.5-VL-3B, a stronger and more recent model. After training on 65K COMPACT data (with 32K compositional tuning data samples), we observe consistent gains: MMVet (59.22 to 60.46) and MMStar (55.80 to 56.05). Although Qwen's training data is not publicly available, these results provide evidence that explicit modeling of complexity using the k-value generalizes to other model architectures and scales.
>
> **W3.** Thank you for the suggestion. We conducted additional experiments comparing COMPACT to a comprehensive set of data reduction, coreset selection, and data pruning methods: EL2N, Perplexity, SemDeDup, D2-Pruning, Self-Sup, and Self-Filter, and ICONS (already reported; Tab. 2 in the paper). As shown in the table below, COMPACT achieves 100.18\% relative performance compared to the full LLaVA-665K dataset, and outperforms all the other newly added baseline methods. These results validate that COMPACT's complexity-aware curation strategy provides consistent gains over traditional data reduction methods that focus primarily on sample difficulty or diversity without considering sample complexity. We will include this expanded comparison in the revision.
>
> | Recipe | # Data | InfoVQA | SeedBench2Plus | MME | TextVQA | MM-Vet | CV-Bench | MMStar | LLaVA-W | Rel. (%) |
> |--------|--------|---------|----------------|-----|---------|--------|----------|--------|---------|----------|
> | LLaVA-665K | 665K | 20.80 | 41.72 | **1478.48** | **46.99** | 29.22 | **60.92** | 35.11 | **68.50** | 100.00 |
> | Random | 65K | 20.05 | 41.85 | 1327.70 | 42.88 | 30.46 | 54.71 | 34.13 | 64.30 | 95.38 |
> | EL2N | 65K | 20.52 | 42.95 | 1378.58 | 42.41 | **33.53** | 50.92 | 33.82 | 66.40 | 97.09 |
> | Perplexity | 65K | 20.46 | 41.90 | 1375.32 | 42.95 | 30.32 | 52.72 | 33.47 | 68.40 | 96.09 |
> | SemDeDup | 65K | 20.54 | **43.96** | 1431.36 | 42.71 | 28.39 | 42.24 | 34.18 | 62.10 | 93.31 |
> | D2-Pruning | 65K | 20.90 | 43.70 | 1343.34 | 41.82 | 31.61 | 48.49 | **36.63** | 68.40 | 97.13 |
> | Self-Sup | 65K | 20.61 | 42.51 | 1434.30 | 42.68 | 30.18 | 54.30 | 34.33 | 61.20 | 96.04 |
> | Self-Filter | 65K | 19.99 | 41.33 | 1290.34 | 36.59 | 28.21 | 45.17 | 33.67 | 66.40 | 90.51 |
> | ICONS | 65K | 21.0 | 42.03 | 1402.75 | 43.12 | 31.23 | 55.96 | 35.96 | 61.8 | 97.47 |
> | COMPACT (ours) | 65K | **23.68** | 43.13 | 1379.94 | 44.37 | 31.74 | 55.28 | 36.13 | 64.50 | **100.18** |
> | COMPACT (ours, qwen3) | 65K | 22.97 | 42.12 | 1370.76 | 43.07 | 30.78 | 56.38 | 34.04 | 65.70 | 98.31 |

---

> > ### Author Response · Authors · 2025-11-27
> >
> > **Q1.** We thank the reviewer for this important question. To address this concern, we conducted a token-level analysis comparing our COMPACT generated compositional tuning data (32k samples) with an equivalent-sized LLaVA baseline (32k samples). The results demonstrate that COMPACT achieves substantial token efficiency despite incorporating multiple atomic questions. Our analysis reveals that COMPACT uses 104.87 tokens/samples (3.36M total tokens) compared to LLaVA's 197.42 tokens/samples (6.32M total tokens), achieving a 46.88\% reduction (92.55 fewer tokens per entry). Specifically, COMPACT's input (question) tokens average $12.83 \pm 4.18$ (31\% shorter than LLaVA's $16.85 \pm 25.49$), and answer tokens average $1.70 \pm 0.90$ (92\% shorter than LLaVA's $21.74 \pm 73.38$). The token reduction translates directly to faster training ($\sim$47\% fewer tokens to process), lower storage usage (shorter sequences), and better data efficiency (more focused atomic capabilities per token).
> >
> > | Metric | COMPACT | LLaVA | Difference |
> > |--------|---------|-------|------------|
> > | Input tokens (mean) | $12.83 \pm 4.18$ | $16.85 \pm 25.49$ | 31\% shorter |
> > | Output tokens (mean) | $1.70 \pm 0.90$ | $21.74 \pm 73.38$ | 92\% shorter |
> > | Tokens per Q\&A turn | 14.53 | 38.59 | 62\% fewer |
> > | Total tokens per entry | 104.87 | 197.42 | 46.88\% reduction |
> >
> > **Q2.** Regarding random data occasionally outperforming the full dataset on individual benchmarks: this is expected due to natural variance in benchmark-specific performance. The relative performance in the aggregate across all benchmarks, where COMPACT consistently outperforms random sampling (Tab. 2 in the paper), is a more reliable measure that reveals the overall trend: higher k-value data facilitates learning.
> >
> > Regarding performance on TextVQA: this is not a weakness but rather evidence that COMPACT successfully avoids dataset-specific overfitting. LLaVA-665K dedicates at least 102K samples (15\% of the dataset) specifically to OCR tasks (OCRVQA and TextCaps), with TextCaps using the same images as TextVQA. This creates a form of implicit test set leakage, where the model is trained on the same images and tasks that it is evaluated on. In contrast, COMPACT samples atomic capabilities uniformly without tailoring to specific benchmark distributions which prevents benchmark-specific optimization. The fact that COMPACT achieves 100.18\% relative performance overall while using only 10\% of the data demonstrates that our approach successfully improves visual reasoning. If a specific application requires strong OCR performance, practitioners can easily adjust the capability sampling distribution to emphasize text recognition, which is precisely the flexibility that COMPACT's framework enables.
> >
> > **Q3.** We provide the average k-value of each benchmark in the table below. As shown in Tab.2 in the paper, COMPACT improves across multimodal benchmarks with various $\bar{k}$ values.
> >
> > | Benchmark | InfoVQA | SeedBench2Plus | MME | TextVQA | MM-Vet | CV-Bench | MMStar | LLaVA-W |
> > |-----------|---------|----------------|-----|---------|--------|----------|--------|---------|
> > | $\bar{k}$ | 0.34 | 1.11 | 1.16 | 1.19 | 1.24 | 1.33 | 1.40 | 3.05 |

---

### Official Review · Reviewer_MPVR · 2025-11-05

**Soundness:** 3
**Presentation:** 3
**Contribution:** 3
**Rating:** 4
**Confidence:** 4

**Summary:**

The main motivation is to compress multiple capabilities into a smaller number of data samples to increase sample efficiency, doing more with less data sets that compose multiple atomic capabilities into one.

**Strengths:**

there are clear taxonomy of capabilities
there are indeed clear gains over the llava-665k datasets where it was not principally constructed.

**Weaknesses:**

see the question

**Questions:**

I think this make sense, but the random baseline is very honest and seem to also suggest that using only 49k out of 665K pretty similar to the COMPACT setup, realistically, SFT is pretty light weight

I would think about improving this work via framing as improving existing answer quality than just data effiency, like many atomic and subjective task here could be used to double check the quality of answers, see if they are correct, or use them in capability-specific abiliations to try to see what task are driving most gains. I would be surprised if color (which seem relatively easy) drive much of the gain

---

> ### Author Response · Authors · 2025-11-26
>
> We thank the reviewer for this valuable feedback and we address the concerns below.
>
> **Q1: Random baseline performance**
>
> Thank you for the observation. While the 49K random baseline performance is relatively high, Tab. 3 in the paper shows that the performance improvement from COMPACT is non-trivial. With 49K samples, the random baseline achieves 96.28% relative performance while COMPACT achieves 98.83%. We can isolate the role of the k-value by comparing COMPACT with COMPACT_llava, whose k-values exactly match the random baseline. COMPACT_llava achieves 97.55%, indicating that roughly half of COMPACT's performance gain can be attributed to increasing the k-value of the training data (ln. 327 in the paper). Furthermore, Fig. 3 in the paper shows that COMPACT is scalable, as its performance on multimodal benchmarks increases as the size of the compositional tuning data grows.
>
> **Q2: Framing as answer quality, and capability-specific ablations**
>
> Thank you for this thoughtful suggestion. The main problem that we aim to address with COMPACT is the continuous scaling of visual instruction tuning data for training MLLMs (ln. 033 in the paper), driven by the emphasis on quantity over quality of the training samples (ln. 038 in the paper). Therefore, we present COMPACT as a data recipe for efficient training from the data budget perspective. We agree that various applications of COMPACT are possible based on its compositional design (Fig. 2 in the paper) and its effectiveness (Tab. 2 in the paper), including the reviewer's suggestion on using COMPACT to "improve existing answer quality". However, the quality of the answer is already reflected as part of the "comprehensive" (reviewer Ak1x, reviewer YsZe) evaluation process. We argue that our current narrative focusing on "data efficiency" is more holistic and central to the main problem.
>
> In order to understand "which tasks are driving the most gains", we conduct a leave-one-out experiment where we remove each atomic capability from COMPACT during data generation. The table below shows the average drop in relative performance across multimodal benchmarks (benchmarks are discussed in Section 4.1 in the paper) when we ablate each atomic capability. The results indicate that all atomic capabilities contribute non-trivially to COMPACT's performance.
>
> | Atomic Capability | Rel. Performance Drop |
> |-------------------|-----------------------|
> | w/o scene understanding | -5.2% |
> | w/o spatial relationship | -4.9% |
> | w/o text recognition | -4.7% |
> | w/o object recognition | -4.0% |
> | w/o color | -3.7% |
> | w/o counting | -3.3% |
> | w/o object interaction | -3.2% |
> | w/o spatial recognition | -3.1% |
> | w/o action recognition | -2.1% |
> | w/o shape | -0.7% |
>
> *Average drop in relative performance as a result of ablating each atomic capability during 16K compositional tuning data generation in COMPACT.*
>
> We also provide a per-category performance breakdown on MMStar to analyze where the performance improvement comes from. The table below shows broad improvements across categories including science & technology (diagram and chart understanding) and instance reasoning (perception and relation understanding), suggesting that the gains are not limited to a single atomic capability such as "color" as the reviewer hypothesized.
>
> | Method | Data | Coarse Perception | Fine-grained Perception | Instance Reasoning | Logical Reasoning | Math | Science & Technology |
> |--------|------|-------------------|-------------------------|--------------------|--------------------|------|---------------------|
> | Random | 49K | 59.5 | 28.9 | 38.1 | 29.9 | 28.7 | 25.2 |
> | COMPACT | 49K | 61.6 | 28.9 | 41.2 | 31.8 | 28.1 | 27.7 |
> | **Improvement** | | **+3.5%** | **0%** | **+8.1%** | **+6.4%** | **-2.1%** | **+9.9%** |
>
> *Performance of COMPACT and random LLaVA subset on each category in MMStar. COMPACT data has 33K VIT data and 16K compositional tuning data.*

---

### Official Review · Reviewer_Ak1x · 2025-11-05

**Soundness:** 4
**Presentation:** 3
**Contribution:** 3
**Rating:** 6
**Confidence:** 3

**Summary:**

This paper presents COMPACT (COMPositional Atomic-to-Complex Visual Capability Tuning), a new data recipe for visual instruction tuning (VIT) in multimodal large language models (MLLMs). COMPACT introduces the idea of compositional complexity, where each training sample is constructed by combining multiple atomic visual capabilities (e.g., object recognition, spatial reasoning, color, shape). By controlling the number of combined capabilities (“k-value”), the method generates more information-dense and complex questions using Gemini-2.0-Flash, leading to significant data efficiency gains. With only 10% of LLaVA-665K data, COMPACT achieves 100.2% of the full-scale performance across benchmarks such as MM-Vet and MMStar, highlighting the benefit of complexity-aware data curation for MLLMs.

**Strengths:**

- Conceptually innovative: Introduces a clear and quantifiable notion of compositional complexity in VIT data, shifting focus from scale to information density.

- Strong empirical evidence: Matches or exceeds full-data performance with one-tenth of the samples, demonstrating outstanding data efficiency.

- Thorough evaluation: Comprehensive experiments across major multimodal benchmarks and detailed ablations on complexity levels (k-values) and instruction-tuning ratios.

- The paper is well-organized, with transparent methodology, taxonomy of atomic capabilities, and plans to release the dataset.

**Weaknesses:**

- Dependency on proprietary models: The reliance on Gemini-2.0-Flash for both question generation and verification is a significant limitation for reproducibility and may introduce unknown biases. The cost ($86.5 for 32K samples) could also be prohibitive for scaling to larger datasets. An analysis using open-source alternatives would strengthen the work.

- While the 10 atomic capabilities are well-defined, the paper acknowledges they are "not expected to be completely orthogonal" but provides limited justification for this specific set. The correlation analysis (Fig. 8) suggests substantial dependencies, yet the implications for the k-value metric are not fully explored. How does correlation between capabilities affect the actual complexity?

- Limited scope of evaluation: The focus is exclusively on vision-centric tasks. The poor performance on knowledge-intensive benchmarks (Table 9) suggests the approach may not generalize to domains requiring external knowledge or reasoning beyond perceptual capabilities. This limits the claim of addressing "general visual reasoning."


- Verification process clarity: The quality verification step (Step 3) uses confidence thresholds and word overlap metrics, but the paper provides limited analysis of failure modes or how often verification rejects generated samples. More transparency about the quality control process would be helpful.

**Questions:**

The "natural integration" requirement for multi-capability questions is somewhat subjective and relies on the LLM's interpretation

Zero-capability samples (0.9% of LLaVA-665K) are interesting but receive minimal discussion

---

> ### Author Response · Authors · 2025-11-26
>
> **W1: Dependency on proprietary models**
>
> Thank you for this suggestion. We have conducted additional experiments using the open-source Qwen3-VL-4B-Instruct model with the same COMPACT generation approach. As shown in the table below, COMPACT (ours, qwen3) with 65K samples achieves 98.31\% relative performance compared to the full LLaVA-665K dataset. While the Gemini-generated version performs slightly better (100.18\%), the Qwen3-generated version still outperforms all other data reduction baselines. As per Reviewer jUJz's suggestion on additional data reduction experiments, we provide additional baseline comparisons, and COMPACT (ours, qwen3) outperforms all of them: ICONS (97.47\%), D2-Pruning (97.13\%), EL2N (97.09\%), Perplexity (96.09\%), Self-Sup (96.04\%), Random (95.38\%), SemDeDup (93.31\%), and Self-Filter (90.51\%). This shows that the benefits of increasing the k-value of the training data holds across different data generators. Notably, COMPACT (ours, qwen3) achieves competitive performance on most benchmarks, with particularly strong results on InfoVQA (22.97), CV-Bench (56.38), and LLaVA-W (65.70). We will include these results in the revision and release all generated data and prompts to ensure full reproducibility.
>
> | Recipe | # Data | InfoVQA | SeedBench2Plus | MME | TextVQA | MM-Vet | CV-Bench | MMStar | LLaVA-W | Rel. (%) |
> |--------|--------|---------|----------------|-----|---------|--------|----------|--------|---------|----------|
> | LLaVA-665K | 665K | 20.80 | 41.72 | 1478.48 | 46.99 | 29.22 | 60.92 | 35.11 | 68.50 | 100.00 |
> | Random | 65K | 20.05 | 41.85 | 1327.70 | 42.88 | 30.46 | 54.71 | 34.13 | 64.30 | 95.38 |
> | EL2N | 65K | 20.52 | 42.95 | 1378.58 | 42.41 | 33.53 | 50.92 | 33.82 | 66.40 | 97.09 |
> | Perplexity | 65K | 20.46 | 41.90 | 1375.32 | 42.95 | 30.32 | 52.72 | 33.47 | 68.40 | 96.09 |
> | SemDeDup | 65K | 20.54 | 43.96 | 1431.36 | 42.71 | 28.39 | 42.24 | 34.18 | 62.10 | 93.31 |
> | D2-Pruning | 65K | 20.90 | 43.70 | 1343.34 | 41.82 | 31.61 | 48.49 | 36.63 | 68.40 | 97.13 |
> | Self-Sup | 65K | 20.61 | 42.51 | 1434.30 | 42.68 | 30.18 | 54.30 | 34.33 | 61.20 | 96.04 |
> | Self-Filter | 65K | 19.99 | 41.33 | 1290.34 | 36.59 | 28.21 | 45.17 | 33.67 | 66.40 | 90.51 |
> | ICONS | 65K | 21.0 | 42.03 | 1402.75 | 43.12 | 31.23 | 55.96 | 35.96 | 61.8 | 97.47 |
> | COMPACT (ours) | 65K | 23.68 | 43.13 | 1379.94 | 44.37 | 31.74 | 55.28 | 36.13 | 64.50 | 100.18 |
> | COMPACT (ours, qwen3) | 65K | 22.97 | 42.12 | 1370.76 | 43.07 | 30.78 | 56.38 | 34.04 | 65.70 | 98.31 |
>
> **W2: Atomic capabilities justification and correlation**
>
> We appreciate this important question. The selection of our 10 atomic capabilities was motivated by two goals: (1) covering fundamental vision-centric skills necessary for multimodal reasoning, and (2) ensuring each capability is sufficiently distinct to enable meaningful combinations. We acknowledge that perfect orthogonality is neither achievable nor necessary, and in fact, real-world visual reasoning naturally involves overlapping skills. For example, spatial reasoning often co-occurs with object recognition, and color attribution frequently appears with shape. The correlation analysis in Fig. 8 in the Appendix reveals these natural dependencies, which we view as a feature rather than a limitation of our taxonomy.
>
> Regarding the k-value metric: it is a heuristic we introduce in an effort to quantify the complexity of a question in an interpretable way, and the correlations do not undermine its validity as a complexity measure. Our empirical validation (on the TextVQA benchmark) in the table below shows that higher k-values consistently correspond to lower model performance, demonstrating that the k-value captures meaningful task difficulty regardless of capability correlations. The key insight is that combining multiple capabilities, even correlated ones, requires the model to integrate more visual information, which increases information density and learning efficiency.
>
> |  | k = 1 | k = 2 | k = 3 |
> |---------|---------|---------|---------|
> | Density | 73.6% | 19.1% | 2.2% |
> | LLaVA-v1.5-7B | 46.4 | 45.6 | 39.8 |
>
> *Note: k≥4 is excluded as it contains fewer than 10 samples. Approximately 5% of the questions are k=0, which are ambiguous and do not strictly refer to the image (e.g., "Are jordan products produced in jordan?").*

---

> > ### Author Response · Authors · 2025-11-26
> >
> > **W3: Limited scope of evaluation**
> >
> > We would like to clarify the intentional scope of COMPACT. Our primary focus on vision-centric capabilities is motivated by the specific role of visual instruction tuning in the MLLM training pipeline. VIT data, including LLaVA-665K which serves as our baseline, is designed to train the models to effectively utilize visual information and ground their responses in image content. In contrast, knowledge-intensive capabilities (e.g., world knowledge, commonsense reasoning, domain-specific facts) are primarily learned during the pre-training stage through large-scale text corpora and image-text pairs.
> >
> > Given this context, COMPACT's emphasis on perceptual capabilities (recognition, attribution, spatial relations) aligns with the core objective of VIT: instruction following grounded in visual content. By focusing on vision-centric skills, COMPACT encourages models to extract and compose rich visual information from images when responding to instructions. Including knowledge-intensive capabilities in our atomic taxonomy would make it difficult to isolate the effect of compositional complexity on visual learning efficiency. Moreover, our category-specific analysis on MMStar in the table below shows that COMPACT does improve performance on logical reasoning (+6.4%) and science & technology tasks (+9.9%), suggesting that stronger visual grounding can transfer to tasks requiring reasoning beyond pure perception. The modest performance on knowledge-intensive benchmarks (Table 9 in the paper) is expected given our design choices and does not undermine COMPACT's effectiveness for improving VIT efficiency. Extending COMPACT to non-perceptual capabilities is promising future work that would require incorporating external knowledge sources beyond what is visually present in images.
> >
> > | Method | Data | Coarse Perception | Fine-grained Perception | Instance Reasoning | Logical Reasoning | Math | Science & Technology |
> > |--------|------|-------------------|-------------------------|--------------------|--------------------|------|---------------------|
> > | Random | 49K | 59.5 | 28.9 | 38.1 | 29.9 | 28.7 | 25.2 |
> > | COMPACT | 49K | 61.6 | 28.9 | 41.2 | 31.8 | 28.1 | 27.7 |
> > | **Improvement** | | **+3.5%** | **0%** | **+8.1%** | **+6.4%** | **-2.1%** | **+9.9%** |
> >
> > *Performance of COMPACT and random LLaVA subset on each category in MMStar. COMPACT data has 33K VIT data and 16K compositional tuning data.*

---

> > > ### Author Response · Authors · 2025-11-26
> > >
> > > **W4: Verification process clarity**
> > >
> > > Thank you for this suggestion. We provide additional details about our quality verification process and its effectiveness. Our verification step (Step 3) employs multiple filtering mechanisms: (1) confidence score threshold (70%), filtering questions where the VLM's self-assessed confidence in the generated Q&A pair is below this threshold; (2) word overlap check (60% threshold), discarding questions that share more than 60% of words with previously accepted questions for the same image to ensure diversity; (3) uninformative answer filtering, removing responses like "unknown", "not visible", "yes", or "no" that do not provide meaningful supervision; and (4) capability verification, prompting the VLM to analyze whether each question indeed requires the k_gen specified capabilities. The generation and verification process repeats iteratively until we collect 2-3 high-quality conversations per k_gen for each image or reach a maximum of 10 verification attempts.
> > >
> > > To quantify the effectiveness of our quality control, we conducted a controlled toy experiment generating questions for k=1, 2, 3 on a sample of images. Our multi-stage filtering rejected approximately 21% of generated questions, distributed across four primary failure modes: (1) low confidence scores (10% in our sample), questions that the VLM cannot answer confidently, often due to ambiguous phrasing or requiring information not present in the image; (2) uninformative responses (12.5% of rejections), such as "What text is visible on the wooden tray?" answered with "None", which fail to provide meaningful visual supervision; (3) high word overlap (40% of rejections), near-duplicate questions for the same image, such as generating both "What is the color of the bench in the image?" and "What is the color of the bench located in the center of the scene?" with 70% word overlap; and (4) capability mismatch (37.5% of rejections), questions that do not naturally integrate the specified capabilities, such as asking "What object is present in the image without any action being performed?" for k=2 with action_recognition and object_recognition, where the question only requires object recognition (the VLM correctly identified only 1 of 2 expected capabilities). Notably, capability mismatch becomes increasingly important as k increases, accounting for 0%, 66.7%, and 50% of rejections for k=1, 2, 3 respectively, demonstrating that our verification successfully filters questions that artificially force capability combinations. This multi-stage quality control ensures that our training data maintains both high quality and natural capability integration.

---

> > > > ### Author Response · Authors · 2025-11-26
> > > >
> > > > **Q1: Natural integration subjectivity**
> > > >
> > > > We agree that "natural integration" involves some subjectivity, which is why we implement the quality verification step (Step 3) to operationalize this requirement. The verification uses the VLM's confidence score (threshold ≥ 70%) as a proxy for whether the question naturally integrates the required capabilities given the image content. Additionally, we prompt the VLM to perform capability verification, analyzing whether each question indeed requires the k_gen specified capabilities (Section 3.2 in the paper). While this relies on the VLM's interpretation, it provides a consistent and scalable quality control mechanism. Our prompt design enforces that questions must integrate the specified capabilities naturally without using conjunctions to simply conjoin single-capability questions (e.g., "What color is the object on the left?" naturally combines color and spatial reasoning, whereas "What color is the object? Is it on the left?" would be unnatural).
> > > >
> > > > **Q2: Zero-capability samples**
> > > >
> > > > Thank you for highlighting this. Zero-capability samples (k=0) are questions that do not require visual information from the image to answer, such as "What country is associated with this brand?" or "What is the capital of France?" These questions can be answered using world knowledge or text in the question alone, without examining the image. We found 0.9% of LLaVA-665K falls into this category, which we consider problematic for visual instruction tuning as they do not encourage visual grounding. In COMPACT, we explicitly avoid generating k=0 samples by design, our data generation pipeline requires at least one atomic visual capability (k≥1) for every question. This ensures that all training samples require meaningful visual reasoning. We will expand the discussion of k=0 samples in the revision to clarify their implications for data quality and why we exclude them from COMPACT.

---

### Official Review · Reviewer_Nn4a · 2025-11-05

**Soundness:** 3
**Presentation:** 3
**Contribution:** 3
**Rating:** 8
**Confidence:** 3

**Summary:**

COMPACT (COMPositional Atomic-to-Complex Visual Capability Tuning) introduces a method for generating complex, information-dense Visual Instruction Tuning (VIT) datasets by combining multiple atomic visual capabilities (e.g., color, spatial reasoning, object recognition) into single training examples. This complexity-aware curation improves data efficiency -- achieving 100.2% of full LLaVA-665K performance using only 10% of the data, with notable gains on complex multimodal benchmarks like MM-Vet and MMStar.

**Strengths:**

- The concept of compositional complexity (k-value) as a controllable metric for VIT data curation is intuitive and well-grounded. The exploratory experiment (Figure 1) effectively demonstrates that increasing k improves performance.

- Strong improvements on compositional tasks with 90 % less data; convincing scaling curves and ablations.

- The paper includes thorough ablations examining complexity ranges, instruction tuning ratios, and complexity distributions. The analysis of LLaVA-665K's complexity distribution (mean k=1.95) provides valuable insights.

- Well-analyzed: Breaks down atomic capabilities, evaluates complexity distributions, and studies instruction-format mixing.

**Weaknesses:**

- Atomic Capability Definition:
  - The taxonomy appears somewhat arbitrary (why these 10 capabilities specifically?)
  - Capabilities are acknowledged as non-orthogonal (Figure 8), undermining the "atomic" framing
  - Object recognition is implicitly assumed in most questions (Figure 7), suggesting the capabilities may not be properly decomposable

- Evaluation scope: Only one base model (LLaVA-v1.5-7B-LoRA); unclear generality to other architectures or scales.

- Error Analysis:
  - Qualitative examples (Figure 11) cherry-pick favorable cases without systematic error analysis
  - Insufficent analysis of failure modes or systematic biases in generated data

**Questions:**

- How sensitive are results to the choice of data generator? Have you experimented with other VLMs (e.g., GPT-4V, LLaVA-NeXT)?

- Is there evidence that naturally occurring questions follow a certain k-distribution? How does COMPACT's distribution compare?

- The paper conflates "compositional complexity" with "task complexity" without clearly distinguishing them.


Minor:
- Figure 3: are you sure its in log scale?

---

> ### Author Response · Authors · 2025-11-27
>
> We sincerely thank the reviewer for the encouraging feedback and valuable suggestions!
>
> **W1: Atomic Capability Definition.** Regarding the selection of atomic capabilities: we selected the atomic capabilities (Tab.1 in the paper) based on two key considerations. First, our goal is to encourage the utilization of rich visual information in the image during training. Therefore, we incorporate vision-centric capabilities that can also be found in existing VIT data curation methods. Second, we need to choose from the capabilities that are represented in the LLaVA-665K dataset, our primary training baseline, for fair comparison. If we include capabilities such as math and coding that are not present in LLaVA-665K, and then evaluate on benchmarks that require these skills, we cannot explain COMPACT's performance gains solely using the k-value. The original LLaVA-665K data curation process lays out 8 of the 10 atomic capabilities in COMPACT; the two others (object interaction and shape attribution) we identified from manually inspecting the dataset. We acknowledge that alternative taxonomies are possible depending on specific training goals, and we view our 10 atomic capabilities as a practical starting point rather than a definitive set.
>
> Regarding the non-orthogonality of capabilities: we agree that perfect orthogonality is neither achievable nor necessary for our framework. Real-world visual reasoning naturally involves overlapping skills—for example, spatial reasoning often co-occurs with object recognition. The correlation analysis in Figure 8 reveals these natural dependencies, which we view as a feature rather than a limitation. The key insight is that combining multiple capabilities, even correlated ones, requires the model to integrate more visual information, which increases information density.
>
> Regarding the implicit assumption of object recognition: we acknowledge this observation from Figure 7. Object recognition serves as a foundational capability that often appears alongside other skills, similar to how reading comprehension is fundamental to many NLP tasks. Rather than undermining decomposability, this reflects the hierarchical nature of visual reasoning where basic perception enables higher-level reasoning.
>
> **W2: Evaluation scope.** We appreciate this feedback. The LLaVA-665K VIT dataset and the LLaVA-v1.5-7B-LoRA model is a reasonable and cost-effective setting to test COMPACT. In order to show that complexity-aware data curation with COMPACT works on stronger models, we conducted a new experiment where we evaluate Qwen2.5-VL-3B on MMVet and MMStar after training on COMPACT data. We observe consistent gains: MMVet (59.22 to 60.46) and MMStar (55.80 to 56.05). Although Qwen's training data is not publicly available, we provide evidence that explicit modeling of complexity using the k-value generalizes to other models.
>
> **W3: Error Analysis.** We conducted a systematic analysis of failure modes and biases in our generated data. To quantify the effectiveness of our quality control, we performed a controlled experiment generating questions for $k_{gen}=1,2,3$ on a subset of LLaVA-665K images. Our multi-stage filtering rejected approximately 21\% of generated questions across four primary failure modes: (1) low confidence scores (10\%), (2) uninformative responses (12.5\%), (3) high word overlap/near-duplicates (40\%), and (4) capability mismatch (37.5\%). Notably, capability mismatch becomes increasingly important as $k$ increases, accounting for 0\%, 50\%, and 66.7\% of rejections for $k=1, 2, 3$ respectively, demonstrating that our verification successfully filters questions that artificially force capability combinations.
>
> Common failure patterns in the remaining data include: (1) Overly complex k=3 questions that are difficult even for humans to answer, (2) Questions where the VLM generator misidentifies image content leading to incorrect ground-truth answers (mitigated by our verification step but not eliminated), (3) Spatial reasoning errors when objects are ambiguously positioned, and (4) Color/attribute questions for small or partially occluded objects. However, COMPACT achieves consistent improvements across diverse benchmarks with different evaluation protocols, suggesting that these biases do not critically harm generalization.

---

> > ### Author Response · Authors · 2025-11-27
> >
> > **Q1.** We conducted experiments with Qwen3-VL-4B-Instruct as an alternative open-source data generator. As shown in the table below, COMPACT (ours, qwen3) achieves 98.31\% relative performance compared to the full LLaVA-665K dataset, which is competitive with our Gemini-based approach (100.18\%) and substantially better than random baseline (95.38\%). The consistent improvements across both closed-source (Gemini-2.0-Flash) and open-source (Qwen3-VL) generators suggest that COMPACT is robust to generator choice. We have not yet experimented with GPT-4V or LLaVA-NeXT, but we expect similar results given the consistency observed so far. The key requirement is that the generator can reliably produce questions that naturally integrate multiple capabilities and verify answer quality, which most capable VLMs can achieve.
> >
> > | Recipe | # Data | InfoVQA | SeedBench2Plus | MME | TextVQA | MM-Vet | CV-Bench | MMStar | LLaVA-W | Rel. (%) |
> > |--------|--------|---------|----------------|-----|---------|--------|----------|--------|---------|----------|
> > | LLaVA-665K | 665K | 20.80 | 41.72 | **1478.48** | **46.99** | 29.22 | **60.92** | 35.11 | **68.50** | 100.00 |
> > | Random | 65K | 20.05 | 41.85 | 1327.70 | 42.88 | 30.46 | 54.71 | 34.13 | 64.30 | 95.38 |
> > | EL2N | 65K | 20.52 | 42.95 | 1378.58 | 42.41 | **33.53** | 50.92 | 33.82 | 66.40 | 97.09 |
> > | Perplexity | 65K | 20.46 | 41.90 | 1375.32 | 42.95 | 30.32 | 52.72 | 33.47 | 68.40 | 96.09 |
> > | SemDeDup | 65K | 20.54 | **43.96** | 1431.36 | 42.71 | 28.39 | 42.24 | 34.18 | 62.10 | 93.31 |
> > | D2-Pruning | 65K | 20.90 | 43.70 | 1343.34 | 41.82 | 31.61 | 48.49 | **36.63** | 68.40 | 97.13 |
> > | Self-Sup | 65K | 20.61 | 42.51 | 1434.30 | 42.68 | 30.18 | 54.30 | 34.33 | 61.20 | 96.04 |
> > | Self-Filter | 65K | 19.99 | 41.33 | 1290.34 | 36.59 | 28.21 | 45.17 | 33.67 | 66.40 | 90.51 |
> > | ICONS | 65K | 21.0 | 42.03 | 1402.75 | 43.12 | 31.23 | 55.96 | 35.96 | 61.8 | 97.47 |
> > | COMPACT (ours) | 65K | **23.68** | 43.13 | 1379.94 | 44.37 | 31.74 | 55.28 | 36.13 | 64.50 | **100.18** |
> > | COMPACT (ours, qwen3) | 65K | 22.97 | 42.12 | 1370.76 | 43.07 | 30.78 | 56.38 | 34.04 | 65.70 | 98.31 |
> >
> >
> > **Q2.** We note that the concept of ''naturally occurring questions'' is challenging to define in general, as there is no established ground truth for what constitutes a ''natural'' k-distribution in the context of visual question answering. However, we can characterize the distribution of k-value in existing VIT datasets to understand current practices. In section 4.3 of the paper, we analyzed the LLaVA-665K dataset, which serves as our primary baseline. Using Gemini-2.0-Flash, we analyzed 5,400 questions from 1,000 randomly sampled images. The analysis reveals a mean k-value of approximately 1.95 and a mode of k=2, with 77\% of questions requiring two or fewer atomic visual capabilities. In comparison, COMPACT's compositional tuning data achieves a mean k-value of 2.89 and a mode of k=3 (analyzed on 7,200 questions from 1,000 randomly sampled images). This demonstrates that COMPACT successfully shifts the distribution toward higher complexity, creating more information-dense training data while maintaining natural language quality.
> >
> > **Q3.** We use ''compositional complexity" and ''task complexity" to refer to the same concept: the k-value of the question in a visual instruction tuning sample which consists of an image and a question-answer pair. Therefore, ''task complexity" emphasizes the k-value specific to the sample, and ''compositional complexity" emphasizes the compositional nature of the k-value (defined as the number of atomic capabilities to answer the question). We acknowledge that using both terms interchangeably may cause confusion, and we will standardize our terminology to primarily use the ''k-value", and  reserve ''compositional complexity" to when discussing the broader notion of the k-value.
> >
> > **Minor: Figure 3 log scale**
> >
> > Yes, Figure 3 uses a logarithmic scale on the x-axis (number of samples) to better visualize the performance trends across the wide range of data sizes. This allows us to clearly show the performance curves for both COMPACT and baselines across different data budgets.

---

> > > ### Comment · Reviewer_Nn4a · 2025-11-27
> > >
> > > Thank you for engaging with the review. I have updated my confidence score accordingly.

---

> > > > ### Author Response · Authors · 2025-11-29
> > > >
> > > > Thank you so much for the reply, we really appreciate it! We’re motivated by the positive feedback and look forward to AC’s comments.

---

### Official Review · Reviewer_SC5N · 2025-11-06

**Soundness:** 2
**Presentation:** 2
**Contribution:** 2
**Rating:** 4
**Confidence:** 3

**Summary:**

This paper proposes COMPACT, where images are from LLaVA-665K, complex instructions are generated by Gemini, to improve data efficiency in multimodal instruction tuning by generating questions that require combinations of atomic visual capabilities. The task complexity is operationalized by the number of atomic capabilities involved (k-value).
The paper demonstrates that increasing task complexity leads to better use of visual information and yields impressive performance. Experiment results shows that with only 10% of the LLaVA-665K data, COMPACT matches or exceeds the full dataset’s performance across a variety of multimodal benchmarks.

This paper presents a practically useful and empirically strong method. The idea of compositional capability tuning is promising and clearly validated by experiments.
However, conceptual and theoretical foundations remain unclear. Several core concepts, such as task complexity, informativeness, information density, and k-value, are used interchangeably without rigorous justification. The mapping from "number of atomic capabilities" to "actual complexity" is assumed rather than demonstrated. In addition, the capability definitions are hand-picked without theoretical or empirical grounding. The analysis section provides statistics but not mechanism-level explanations. As a result, important questions remain unanswered: Why does COMPACT help? Which capabilities benefit? Why does improvement transfer to tasks outside the covered perceptual abilities?

If supplemented with theoretical evidence, more in-depth analysis and argumentation, this work has the potential to become a very influential contribution, and I will raise your rating.

**Strengths:**

1. "Atomic-to-complex visual capability tuning" is a novel and valuable problem. This paper addresses a real gap in current data curation approaches for MLLM visual instruction tuning. The method is simple yet impactful.
2. The proposed COMPACT dataset demonstrates significantly higher task complexity and achieves 100% full-data performance using only 10% of the original dataset. The authors validate results on 16 complex multimodal tasks, showing clear improvements over baselines.
3. This paper have good motivation via dataset analysis. The paper conducts a detailed study on the complexity distribution of existing datasets. Especially Figure 1, which shows overrepresentation of low-k samples, provides compelling motivation for why higher-complexity samples are needed.
4. The proposed synthetic data recipe is easy to implement and clearly "works in practice". The method directly offers a more data-efficient way to perform visual instruction tuning.

**Weaknesses:**

1. The paper repeatedly uses the terms "task complexity", "informativeness", "effective use of information content", and "k-value" as if they were equivalent. Is informativeness = complexity? Is task complexity = number of atomic capabilities? Why is k=3 more “complex” than k=1 in a meaningful sense?
The paper does not provide a theoretical justification nor an empirical validation for these assumptions. As a result, the k-value appears arbitrary and not a reliable measure of complexity.

2. The atomic capabilities define only basic perceptual skills, making the notion of "complex tasks" overly simplistic. COMPACT’s complexity is defined solely as combinations of perception + attributes + spatial relations. This is a very narrow interpretation of "complexity", and does not align with real-world multimodal complexity, which includes OCR, counting, commonsense, math, reasoning, etc.

3. The choice of the categories seems ad-hoc. Table 1 says

> "We identify 10 atomic capabilities that are necessary for general visual reasoning."

Line 189-190:

> "... but instead provide sufficient coverage of the multimodal task space and to systematically combine tasks of increasing complexities."

We don't understand how you concluded that "the 10 atomic capabilities provide sufficient coverage of the multimodal task space". What evidence supports the conclusion?

The paper does not explain why these are necessary, why others are excluded, and whether the taxonomy is derived from theory, data statistics, or prior work. The capability selection appears subjective and weakens the foundation of the method.

4. No explanation for where the improvement comes from. Benchmarks like MM-Vet and MMStar include tasks requiring OCR, math, logic, world knowledge, far beyond COMPACT’s three perceptual dimensions. Since COMPACT does not train OCR/math/logic abilities, why does it improve these tasks? The paper does not analyze this cross-capability transfer.

5. No breakdown showing whether COMPACT improves perception-only tasks vs. all task categories. Without category-level gains, readers cannot tell whether COMPACT only helps “vision-centric” tasks, or whether simple perceptual compositionality generalizes/transfers to OCR/knowledge/logic. If perceptual complexity generalizes broadly, the claim becomes much stronger. But the paper does not provide the crucial analysis.

6. The paper only provides descriptive correlations, not causal evidence.
The explanation is high-level and speculative. Causal claim "higher complexity $\rightarrow$ higher information density $\rightarrow$ better learning" remains unproven.

7. Training strategy is insufficiently analyzed. The final training mixture is COMPACT + 5% simple LLaVA data. The paper states that both simple and complex samples are necessary, but does not explain why. This likely involves curriculum learning or optimization stability, but the paper provides no analysis or validation.

8. Analysis section is statistics only, not mechanism. The paper largely reports distributions (k-distribution, capability distribution, correlations) but does not explain the mechanism of improvement.
Readers want to know why this works, not just which ablation performs best.

**Questions:**

NA

---

> ### Author Response · Authors · 2025-11-25
>
> Thank you for the careful and insightful review. To summarize, it appears there are three main points to clarify:
>
> 1. The meaning and connection between "task complexity," "informativeness," "k-value." Why do we define k-value as a meaningful quantification of complexity? (weakness 1)
> 2. Selection of atomic capabilities. (weaknesses 2-3)
> 3. Further analysis on where COMPACT's performance improvement comes from, including mechanisms/causal evidence as to why COMPACT works. (weaknesses 4-8)
>
> We will aim to address each of these in detail below. We are also working on a corresponding revision of the manuscript but would greatly appreciate your feedback in the meantime.

---

> > ### Author Response · Authors · 2025-11-25
> >
> > **1. Relationship between task complexity and k-value (weakness 1)**
> >
> > The "k-value" is a heuristic we introduce in an effort to quantify the complexity of a question in an interpretable way. A visual instruction tuning sample is in the form of an image with a question-answer pair, and we define the k-value as the number of atomic capabilities required to answer a given question (ln. 052 in the paper; we comment on our selection of atomic capabilities in the response below). Therefore, "task complexity" is the k-value of the question in the visual instruction tuning sample. We reason that questions with higher k-value require the model to extract and compose more visual evidence in the image, and thus provides a higher information density per sample during training. Our motivational experiment (Fig.1 in the paper) supports this claim by showing that training the model on higher k-value samples leads to better performance on multimodal benchmarks, showing higher informativeness of higher k-value datasets.
> >
> > We run an additional experiment to check whether increasing the k-value indeed makes the task progressively more challenging and hence more "complex." To do so, we analyze TextVQA, a text recognition (OCR) benchmark. Every question in this benchmark contains one key atomic capability (text recognition; Tab.1 in the paper) in addition to potentially others. We split the TextVQA validation set according to the k-value of each question, and report the results of the LLaVA-v1.5-7b model below:
> >
> > |                      | **k = 1** | **k = 2** | **k = 3** |
> > |----------------------|-----------|-----------|-----------|
> > | Density         | 73.6%     | 19.1%     | 2.2%      |
> > | LLaVA-v1.5-7B     | 46.4      | 45.6      | 39.8      |
> >
> > **Table1**: Performance of LLaVA-v1.5-7B on each k-value in TextVQA. $k \geq 4$ is excluded as it contains less than 10 samples. Approximately 5% of the questions are $k=0$, which are ambiguous and does not strictly refer to the image (e.g. Are jordan products produced in jordan?).
> >
> > We find that the LLaVA-v1.5-7B model struggles more as other capabilities (e.g. color, spatial relation understanding) are added to text recognition. This provides further evidence that complexity measured by the k-value is related to the model's ability to process visual information, and that the k-value characterizes the complexity of the question in a meaningful way. **In short, as the k-value increases, overall task difficulty increases.**
> >
> > We report on the TextVQA benchmark because the results are more interpretable. Every question requires a shared atomic capability (text recognition) and has the same answer format, allowing us to isolate the effect of adding capabilities more strictly.

---

> > > ### Author Response · Authors · 2025-11-25
> > >
> > > **2. Selection of atomic capabilities (weakness 2-3)**
> > >
> > > We selected the atomic capabilities (Tab.1 in the paper) based on two key considerations. First, our goal is to encourage the utilization of rich visual information in the image during training. Therefore, we incorporate "perceptual" capabilities that can also be found in existing VIT data curation methods (Tong et al. Cambrian-1, Section 5; Li et al., Eagle 2, Section 2).
> > > With regard to weakness 2, we note that our list of atomic capabilities, which **does include** OCR (text recognition) and counting, is not narrow but rather captures the skills learned from LLaVA-665K comprehensively, as COMPACT achieves broader improvements across various multimodal task categories (see category-specific improvement analysis below).
> > >
> > > Second, we need to choose from the capabilities that are represented in the LLaVA-665K dataset, our primary training baseline, for fair comparison. If we include capabilities such as math and coding that are not present in LLaVA-665K, and then evaluate on benchmarks that require these skills, we cannot explain COMPACT's performance gains solely using the k-value. The original LLaVA-665K data curation process lays out 8 of the 10 atomic capabilities in COMPACT (Liu et al. Visual Instruction Tuning, Section 3); the two others (object interaction and shape attribution) we identified from manually inspecting the dataset. This responds to the question in weakness 3 on how the list of created. We will revise the paper to provide more detailed explanation of our taxonomy construction process.

---

> > > > ### Author Response · Authors · 2025-11-25
> > > >
> > > > **3. Where does COMPACT's performance improvement come from and why does it work? (weakness 4-8)**
> > > >
> > > > We ran some analysis in the original paper (increasing LLaVA-665K k-value distribution in Fig.1, matching LLaVA-665K k-value distribution in Tab.3, ablation on k-value range in Fig.4, ablation on data mixture in Fig.5) but are very happy to conduct additional experiments to better understand where COMPACT's performance improvement comes from. Below we provide category-specific and k-value-specific breakdowns.
> > > >
> > > > **Breakdown of performance by category.** We split the questions in MMStar by category (provided by the dataset) and report category-specific results on the models. Tab.2 shows that COMPACT improves performance across most categories except math, which we exclude from our list of atomic capabilities for fair comparison with training on LLaVA-665K data.  We observe particularly large improvements on science & technology (diagram and chart understanding) and instance reasoning (perception and relation understanding) categories, suggesting that COMPACT broadly generalizes to visual tasks. So, as the reviewer hypothesized in weaknesses 4 and 5, we do observe some degree of "cross-capability transfer". COMPACT improves on tasks that require "logic/knowledge" in combination with strong perception.
> > > >
> > > > | **MMStar**     | **Data** | **coarse perception** | **fine-grained perception** | **instance reasoning** | **logical reasoning** | **math** | **science and technology** |
> > > > |----------------|----------|------------------------|------------------------------|-------------------------|------------------------|----------|-----------------------------|
> > > > | Random        | 49K      | 59.5              | 28.9             | 38.1                    | 29.9                   | 28.7     | 25.2                        |
> > > > | COMPACT    | 49K      | 61.6              | 28.9             | 41.2                    | 31.8                   | 28.1     | 27.7                        |
> > > > | Improvement |         | +3.5%           | 0%             | +8.1%                   | +6.4%             | −2.1%    | +9.9%                 |
> > > >
> > > > **Table 2:** Performance of COMPACT and random LLaVA-665K subset on each category in MMStar. COMPACT data has 33K VIT data and 16K compositional tuning data.
> > > >
> > > > **Higher k-value leads to better learning.** The motivational experiment in Fig.1 of paper provides direct evidence to the "causal claim" mentioned in weakness 6. In Fig.1, we take a random subset of LLaVA-665K (LLaVA) and increase the k-value of each question by regenerating with one additional capability (LLaVA$\_{k+1}$) . We also prepare a control dataset that is regenerated without capability addition (LLaVA$\_{k+0}$). We observe that the overall performance on multimodal benchmarks improve with LLaVA$\_{k+1}$ compared to LLaVA$\_{k+0}$ and LLaVA. After controlling for the images, number of samples, and the regeneration method, higher k-value does lead to higher performance, and thus better learning.
> > > >
> > > > **Breakdown of performance by k-value.** We provide further analysis to explain the "mechanism of improvement" mentioned in weakness 8. We split the questions in MMStar by k-value and measure the performance of three models each trained on three different datasets (k$\_{gen}$=1,k$\_{gen}$=1,2, and k$\_{gen}$=1,2,3; Fig.4 in the paper) with progressively higher k-value distributions. Tab.3 shows that k$\_{gen}$=1,2,3 outperforms others on higher k-value ($k \geq 2$) questions. The improvement is larger when we compare k$\_{gen}$=1,2,3 and k$\_{gen}$=1, as opposed to k$\_{gen}$=1,2,3 and k$\_{gen}$=1,2. These results suggest that increasing the k-value of the training data enables the model to perform well on higher k-value questions in the test dataset.
> > > >
> > > > | **MMStar**                                      | **Data** | **k = 1** | **k = 2** | **k = 3** | **k = 4** |
> > > > |------------------------------------------------|----------|-----------|-----------|-----------|-----------|
> > > > | k$\_{gen}$=1,2,3 vs k$\_{gen}$=1   | 49K      | -0.5%     | +3.6%     | +22.7%    | +33.5%    |
> > > > | k$\_{gen}$=1,2,3 vs k$\_{gen}$=1,2 | 49K      | -1.3%     | +2.6%     | +14.1%    | +9.1%     |
> > > >
> > > > **Table 3:** Performance improvements on each k-value in MMStar by k$\_{gen}$=1,2,3 dataset compared to k$\_{gen}$=1,2,3 and k$\_{gen}$=1,2.
> > > >
> > > > Tab.3 also shows that training on k$\_{gen}$=1,2,3 leads to a small drop in performance on lower k-value ($k=1$) questions, indicating a trade-off between lower and higher k-value regimes. This explains why both simple and complex samples are necessary for training, as pointed out in weakness 7. Furthermore, the 5\% of LLaVA-665K data is necessary for domain adaptation (ln. 083 in the paper). The ablation experiment in Fig.5 of the paper shows that the "simple LLaVA data'' is necessary to format model's responses for various question types, as it includes prompt engineering unlike COMPACT's generated data.
> > > >
> > > > Thank you again for the suggestions!

---

### Author Response · Authors · 2025-12-01
**Thank you!**

We deeply appreciate the Area Chairs taking on this paper under challenging circumstances following the recent ICLR incident. We understand the additional workload this creates for Area Chairs, and to help navigate the substantial volume of feedback across six reviews and our detailed responses, we provide this summary highlighting the key strengths recognized by reviewers, the main concerns raised, and how we have thoroughly addressed each one.


We thank all reviewers for their thoughtful and constructive feedback, and are delighted that they found our work "conceptually innovative" (Reviewer Ak1x), introducing "a novel lens" (Reviewer YsZe) with "a novel data synthesis approach" (Reviewer jUJz) that addresses a "novel and valuable" problem (Reviewer SC5N), our method "intuitive and well-grounded" (Reviewer Nn4a), "simple yet impactful" (Reviewer SC5N), and providing "a principled and measurable axis for dataset construction" (Reviewer YsZe), demonstrating "outstanding data efficiency" (Reviewer Ak1x) with "strong improvements" (Reviewer Nn4a) that "improves finetuning efficiency" (Reviewer jUJz), supported by "strong methodological rigor" (Reviewer YsZe) and "thorough ablations" (Reviewers SC5N, Nn4a, Ak1x), and that the paper is "well-organized" and "transparent" (Reviewers Ak1x, YsZe).

We address the main concerns below and incorporate their feedback into the revision.

**Summary of Revisions**

We incorporate the following changes to the revision (highlighted in blue in the revised manuscript) to help reviewers and ACs focus on key changes; note that we have also made various minor writing improvements, figure adjustments, and clarifications throughout the paper that are not highlighted):

1. Additional generator experiments (Reviewers Nn4a, Ak1x, jUJz, YsZe): Experiments using open-source Qwen3-VL-4B-Instruct as data generator, demonstrating the compositional complexity principle is robust to generator choice.
2. Additional baseline comparisons (Reviewer jUJz): Six additional data reduction baselines (EL2N, Perplexity, SemDeDup, D2-Pruning, Self-Sup, Self-Filter), demonstrating that our complexity-aware approach outperforms traditional data reduction methods.
3. Token-level efficiency analysis (Reviewer jUJz): Analysis showing COMPACT achieves 46.88% token reduction compared to LLaVA-665K, demonstrating efficiency at both sample and token levels.
4. Quality verification analysis (Reviewers Nn4a, Ak1x): Controlled experiment quantifying rejection rates (21%) and detailed examples of failure modes.
5. Category-specific performance analysis (Reviewers SC5N, Ak1x, MPVR, YsZe): Analysis on MMStar showing improvements across multiple categories including logical reasoning (+6.4%) and science & technology (+9.9%), demonstrating visual grounding transfers to reasoning tasks.
6. Leave-one-out capability analysis (Reviewer MPVR): Systematic analysis showing all atomic capabilities contribute non-trivially to performance, with scene understanding, spatial relationship, and text recognition being most impactful.
7. k-value complexity validation (Reviewers SC5N, Ak1x): TextVQA analysis demonstrating higher k-values correspond to lower model performance, validating k-value as a meaningful complexity measure.
8. Technical clarifications & writing improvements (Reviewers SC5N, Nn4a, Ak1x): Detailed explanations of atomic capability taxonomy construction, terminology standardization (k-value, compositional complexity, task complexity), clarified scope of vision-centric focus, and expanded discussion throughout.

We sincerely thank the Area Chairs and all reviewers for their time, effort, and thoughtful engagement with our work, especially under these challenging circumstances. We are grateful for the opportunity to address the concerns and we hope the revisions show our commitment to producing impactful research for the community.

---

### Meta-Review · Area_Chair_NXcp · 2026-01-05

**Summary:**

The paper introduces a novel data recipe for visual instruction tuning (VIT) that enhances sample complexity by combining atomic visual capabilities (e.g., object recognition, spatial reasoning) into single training examples, quantified by a k-value metric. The authors demonstrate that COMPACT achieves 100.2% of the full LLaVA-665K dataset performance using only 10% of the data, with notable gains on complex benchmarks like MM-Vet (+8.6%) and MMStar (+2.9%), supported by rigorous evaluations across eight multimodal benchmarks. Reviewers generally praised the conceptual innovation, strong empirical results, and thorough ablations, with SC5N highlighting the practical utility but questioning the theoretical grounding of k-value and atomic capability selection, while Nn4a appreciated the intuitive complexity metric but noted arbitrary capability definitions and limited error analysis. Ak1x commended the data efficiency but raised concerns about dependency on proprietary models (Gemini) and scope limited to vision-centric tasks, whereas jUJz emphasized the need for comparisons with other data reduction methods and token-level efficiency. YsZe valued the principled approach but suggested expanding capabilities to non-perceptual skills. In rebuttal, the authors addressed these points comprehensively: they conducted additional experiments with open-source models (Qwen3-VL), showing robust performance (98.31% relative score), added six data reduction baselines (e.g., EL2N, SemDeDup) where COMPACT outperformed all (100.18%), provided token-level analysis revealing 46.88% reduction in tokens, and included category-specific breakdowns (e.g., MMStar logical reasoning +6.4%) and leave-one-out capability ablations to demonstrate all atomic capabilities contribute non-trivially. Revisions also clarified taxonomy construction, added failure mode analysis (21% rejection rate), and standardized terminology, strengthening the work's validity and reproducibility. Overall, the paper makes a contribution to data-efficient MLLM training by leveraging compositional complexity, with rebuttals adequately addressing reviewer concerns.

**Reviewer Concerns:**

Addressed Concerns:

* Dependency on Proprietary Models (Reviewers Ak1x, jUJz, YsZe):​ The authors compellingly addressed this by conducting new experiments with the open-source Qwen3-VL model, demonstrating that COMPACT's performance gains (98.31% relative score) are robust and not specific to Gemini, thereby significantly alleviating reproducibility concerns.

* Lack of Baseline Comparisons (Reviewer jUJz):​ This was thoroughly addressed by adding comparisons against six data reduction methods (EL2N, Perplexity, SemDeDup, D2-Pruning, Self-Sup, Self-Filter), showing COMPACT's superiority over these established techniques.

* Token-Level Efficiency (Reviewer jUJz):​ The authors provided a detailed token-level analysis, showing COMPACT achieves a 46.88% reduction in tokens compared to LLaVA data, effectively demonstrating efficiency beyond just sample count.

* Need for Deeper Analysis (Reviewers SC5N, MPVR):​ The rebuttal added significant analysis: a category-specific breakdown on MMStar, a leave-one-out capability ablation showing each capability's contribution, and a validation of the k-value metric on TextVQA. This provides much-needed mechanistic insight into where improvements originate.

* Clarity on Quality Verification (Reviewer Ak1x):​ The authors provided quantitative details on their verification process, including a 21% rejection rate and a breakdown of failure modes, adding transparency.

Outstanding or Partially Addressed Concerns:

* Theoretical Grounding of Atomic Capabilities (Reviewer SC5N):​ While the authors provided empirical justification (coverage of LLaVA-665K capabilities) and pragmatic reasoning (focus on vision-centric skills), the fundamental question regarding the theoretical basis for selecting these specific 10 capabilities and their non-orthogonality remains. This is acknowledged as a conceptual limitation rather than a practical one.

* Scope Limited to Vision-Centric Tasks (Reviewers Ak1x, YsZe):​ This was acknowledged as an intentional design choice to align with the goals of Visual Instruction Tuning. While the authors showed some transfer to reasoning tasks, the concern that COMPACT does not address non-perceptual skills (knowledge, math) is still valid, though it frames the scope for future work rather than a flaw in the current contribution.

* Evaluation on a Single Model/Dataset (Reviewer jUJz):​ Although experiments with Qwen2.5-VL-3B were mentioned, showing a trend, a more comprehensive evaluation across a wider range of model architectures and scales would further strengthen the generalizability claims. The focus on LLaVA-v1.5 and LLaVA-665K remains a limitation in the presented work.

**Reviewer Scores:**

* Reviewer SC5N (Initial score: 4): The rebuttal provided empirical validations, such as k-value complexity analysis on TextVQA and category-specific breakdowns, which partially addressed concerns about theoretical grounding. However, fundamental questions about the atomic capabilities' justification and mechanism-level explanations remain somewhat unresolved. Thus, the score might keep 4, as the practical improvements are clear, but theoretical weaknesses persist.

* Reviewer Nn4a (Initial score: 8): The initial score was already high, indicating strong approval.

* Reviewer Ak1x (Initial score: 6): Key concerns about dependency on proprietary models were addressed with Qwen3-VL experiments, and verification clarity was improved with detailed failure mode analysis.  The score might keep 6.

* Reviewer jUJz (Initial score: 6): The rebuttal directly tackled concerns by adding six baseline comparisons, token-level efficiency data, and open-source model results. This comprehensively addressed issues of reproducibility and baseline gaps. The score might keep 6.

* Reviewer YsZe (Initial score: 6): The use of open-source models and additional analyses on capability generalization partially addressed concerns about model dependency and scope. However, the limitation to vision-centric capabilities remains an intentional design choice, which might not fully satisfy calls for broader applicability. The score might keep 6.

---

### Decision · Program_Chairs · 2026-01-26

Accept (Poster)